# The S2M meteorological and snow cover reanalysis over the French mountainous areas: description and evaluation (1958 - 2021)

Matthieu Vernay[1], Matthieu Lafaysse[1], Diego Monteiro[1], Pascal Hagenmuller[1], Rafife Nheili[1], Raphaëlle Samacoïts[1,2], Deborah Verfaillie[3], and Samuel Morin[1]

[1]Univ. Grenoble Alpes, Université de Toulouse, Météo-France, CNRS, CNRM, Centre d'Études de la Neige, 38000 Grenoble, France
[2]Météo-France, Direction de la Climatologie et des Services Climatiques, Toulouse, France
[3]Earth and life institute, Université Catholique de Louvain, Louvain-La-Neuve, Belgium

**Correspondence:** Matthieu Vernay
(matthieu.vernay@meteo.fr)

**Abstract.** This work introduces the S2M (SAFRAN - SURFEX/ISBA-Crocus - MEPRA) meteorological and snow cover reanalysis in the French Alps, Pyrenees and Corsica, spanning the time period from 1958 to 2021. The simulations are made over elementary areas, referred to as massifs, designed to represent the main drivers of the spatial variability observed in mountain ranges (elevation, slope and aspect). The meteorological reanalysis is performed by the SAFRAN system, which combines information from numerical weather prediction models (ERA-40 reanalysis from 1958 to 2002, ARPEGE from 2002 to 2021) and the best possible set of available in-situ meteorological observations. SAFRAN outputs are used to drive the Crocus detailed snow cover model, which is part of the land surface scheme SURFEX/ISBA. This model chain provides simulations of the evolution of the snow cover, underlying ground, and the associated avalanche hazard using the MEPRA model. This contribution describes and discusses the main climatological characteristics (climatology, variability and trends), and the main limitations of this dataset. We provide a short overview of the scientific applications using this reanalysis in various scientific fields related to meteorological conditions and the snow cover in mountain areas. An evaluation of the skill of S2M is also displayed, in particular through comparison to 665 independent in-situ snow depth observations. Further, we describe the technical handling of this open access data set, available at this address: http://dx.doi.org/10.25326/37#v2020.2. Scientific publications using this dataset must mention in the acknowledgments: "The S2M data are provided by Météo-France - CNRS, CNRM Centre d'Etudes de la Neige, through AERIS" and refer to it as Vernay et al. (2022).

## 1 Introduction

The assessment of fluctuations and long term changes in meteorological and snow cover conditions in mountain regions is critical for many scientific studies and related operational applications (Hock et al., in press). However, the very complex topography of mountains makes the meteorological monitoring of these areas very challenging (Beniston et al., 2018). Numerical modelling based on physical processes allows to extend the information provided by this limited number of observation

stations to wider mountain areas, over longer and uninterrupted time periods, and also to atmospheric and snow cover variables, which cannot be directly observed. Robust assessments of mountain climate evolution is increasingly relying on specific retrospective meteorological analyses (reanalyses) combining a numerical simulation of relevant variables and processes, and

25 past observations. The spatial resolution of existing global reanalyses such as the ECMWF ERA-Interim (79 km, Dee et al., 2011) or its successor ERA-5 (31 km, Hersbach et al., 2019), the NASA MERRA-2 (~50 km, Gelaro et al., 2017) or the Japan Meteorological Agency JRA-55 (~55 km, Kobayashi et al., 2015) is generally too coarse for direct use in mountain regions. For example, Daloz et al. (2020) compared snowfall estimates of these reanalyses over worldwide mountainous areas and showed their limits to capture local orographic enhancements. Such large scale reanalyses usually do not account for precipitation

observations, which is solely output from the model forecasts benefiting from the analysis of other key atmospheric variables (temperature, wind speed etc.). Regional to local reanalyses benefit from a higher spatial resolution, but they often lack key outputs for addressing mountainous regions. For example in France, Gottardi et al. (2012) and Soci et al. (2016) limited their analysis to daily precipitation fields and Caillouet et al. (2019) only considered temperature and evapo-transpiration in addition to daily precipitation.

Reanalyses dedicated to mountain areas have been developed using different methodologies. Margulis et al. (2016) applied a particle batch smoother (Margulis et al., 2015) to produce a reanalysis of snow water equivalent in the Sierra Nevada (USA) over a 30 year period. Bucchignani et al. (2013) used a non-hydrostatic regional climate model at a spatial resolution of 14 km to produce a reanalysis of the meteorological conditions in the Alpine region over the 20th century. Fiddes et al. (2019) developed an ensemble approach to quantify the uncertainties of the combination of a meteorological model and a land surface model,

with a clustering of the simulation points to reduce the computation cost and apply the method at different scales. Olefs et al. (2020) applied the SNOWGRID snow cover model to a climate configuration in order to assess changes in meteorological and snow cover conditions in Austria from 1961 to 2020. However, none of these reanalyses used a multi-layer sophisticated snow cover model model able to describe in details the internal properties of snow on the ground.

Since the 1980s, Météo-France has developed a numerical model chain covering the main French mountain ranges de-

45 signed for operational monitoring and forecasting of snow conditions and avalanche hazard. Initially referred to as SAFRAN-Crocus-MEPRA (SCM, Durand et al., 1999), this model chain simulates both meteorological and snow cover variables, as well as various avalanche hazard diagnostics at various elevations, slopes and aspects for the three main French high elevation mountainous regions (French Alps, Pyrenees and Corsica, see Figure 2). The SAFRAN analysis system (Durand et al., 1993) combines meteorological observations and output from a Numerical Weather Prediction (NWP) model to drive the Crocus

snowpack model (Brun et al., 1989, 1992; Vionnet et al., 2012) with the relevant meteorological variables. Although the initial goal of the system is to provide real-time estimates of snow conditions (Morin et al., 2020), it is also possible to use past data as input to use the SCM chain as a reanalysis tool (Durand et al., 2009a, b) for combined meteorological and snow cover conditions in mountainous areas. This reanalysis has been used to assess the quality of the real-time model chain. Indeed, it provides the simulated variables over a period that is long enough to perform a robust statistic evaluation by comparison with

an independent set of observations of those same variables. It is also a unique source of information concerning past snowpack stability and avalanche hazard, and has been used in a large number of scientific applications in the mountain environment.

This simulation system was later expanded to covering all of mainland France and Corsica for hydrological monitoring and forecasting purpose (SAFRAN-France - ISBA - MODCOU, SIM) (Vidal et al., 2010; Le Moigne et al., 2020), and provided inspiration for a European-scale analysis system (Soci et al., 2016; Morin et al., 2021). However, in this article we only focus on the "original" model chain addressing mountain regions of France.

This paper introduces the latest version of the SCM reanalysis, now referred to as SAFRAN - SURFEX/ISBA-Crocus - MEPRA (S2M). This new version differs from the previous one (Durand et al., 2009a, b) by its temporal extent (15 more years, now spanning 1958-2021), its extension to Corsica and the Pyrenees in addition to the French Alps, and an update of the observations and models involved. For example, Crocus is now fully embedded as a snow cover model of the ISBA land surface model within the SURFEX interface (Lafaysse et al., 2013; Masson et al., 2013). However, the major innovation is that this new dataset is now freely available (see section 3 for the dataset description and section 7 for access informations) for scientific applications using the AERIS portal (http://dx.doi.org/10.25326/37#v2020.2). The first part of this paper describes the input data and model chain used to produce this reanalysis as well as the simulated variables and details on the practical access to the reanalysis dataset. The last section gives an overview of the possible uses of this dataset, with an emphasis on its three main dimensions (spatial, temporal and altitudinal). It also presents an objective assessment of the limitations of this dataset in terms of climate trends, which have never been published until now in spite of its large use in the French scientific community for climate applications. Last, an evaluation of the reanalysis performance is given in terms of total snow depth by comparison to independent observations. Although the S2M dataset is provided and available from 1958 to 2021, all the results of this study are based on the period 1958-2020 of the dataset because the last year was not available at the time it has been carried out.

## 2 Design and main features of the S2M model chain

The S2M reanalysis is the combination of :

- the SAFRAN meteorological analysis, which combines output from a Numerical Weather Prediction model and in situ observations,

- the SURFEX/ISBA-Crocus snow cover model (including MEPRA), which is driven by atmospheric fields from the SAFRAN reanalysis.

### 2.1 Geometry of the S2M reanalysis

The concept of semi-distributed modelling (i.e. Hydrological Response Units) has been widely used since Beven and Kirkby (1979) to discretize a hydrological catchment following the main drivers of the spatial variability of the key processes with an optimal numerical cost (e.g. MacDonald et al. (2010), Fiddes and Gruber (2012), Ajami et al. (2016), Meng et al. (2018)). In mountainous environments, elevation, aspect and slope are known to be the main drivers of the spatial variability of snow energy balance, due to the strong dependence of temperature and radiation on these topographic features. Therefore, topographic

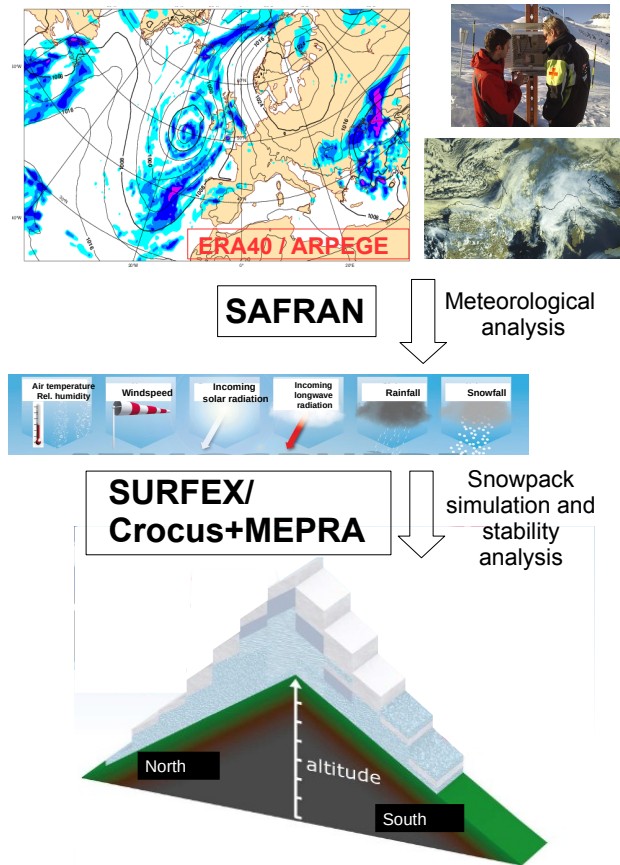

**Figure 1.** Description of the three steps of the reanalysis model chain : 1) NWP model (ERA-40 before 2002, ARPEGE from 2002 onwards) 2) Assimilation and geometry adjustment by SAFRAN 3) Snow cover model SURFEX/ISBA-Crocus, including MEPRA.

classes are the cheapest solution of numerical discretization to represent this variability and is a common modelling choice in alpine hydrology (Lafaysse et al., 2011; Tarasova et al., 2016; Garavaglia et al., 2017), consistently with the pioneer intro-

90 duction of Snow Cover Units by Seidel et al. (1983) and Ehrler et al. (1997) for snow remote sensing. This concept was also chosen by Durand et al. (1999) for the operational application of SAFRAN-Crocus modelling in support of avalanche hazard forecasting. Consistently, the S2M reanalysis results from simulations performed over elementary areas specifically designed to represent the main drivers of the spatial variability in mountain ranges called "massifs" (shapefiles of the different massifs are included in the dataset and a glimpse of the geographic division of the three areas can be seen on Figure 2). A massif

is a conceptual object corresponding to a mountainous area (of about $1000 \, \mathrm{km}^2$ on average) over which the meteorological conditions are considered homogeneous at a given elevation. This hypothesis simplifies the representation of a complex topography by covering the different elevations and aspects of a given massif with a minimum number of representative computation points. An example of this simplification of a real topographic massif is provided in Figure 3. The S2M reanalysis uses a 300 m

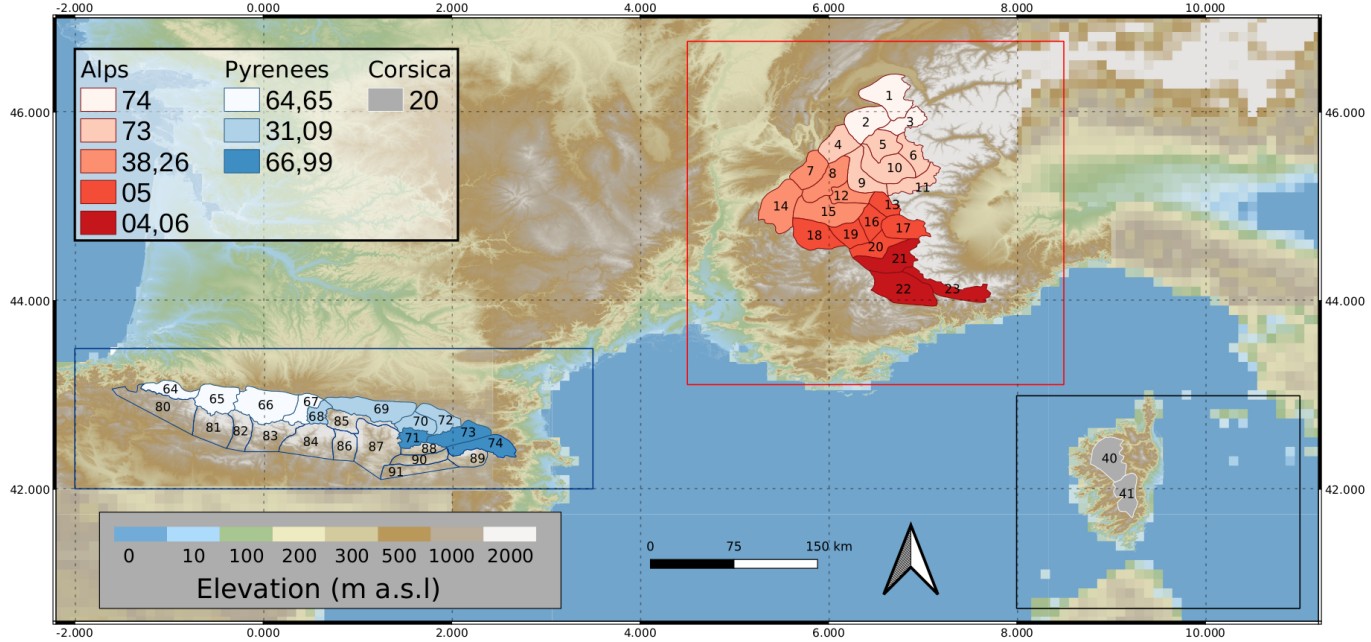

**Figure 2.** Map of the three areas covered by the S2M reanalysis. The rectangles indicate for each area the domain of extraction of the observations and guess from the NWP output. The numbers within the massifs is the one used to identify the massifs in the dataset. The color of the massifs indicates the French administrative department(s) in which the largest part of the massif stands. See Table 8 in Appendix A for more informations.

vertical resolution. The upper (resp. lower) elevation for each massif is defined as the 300 m multiple immediately above the
100 highest point (resp. below the lowest point) of a 50 m digital elevation model from the French National Geographic Institute (IGN) in the considered massif. For each elevation, two different simulations are provided: one simulation only contains flat terrain, and for the other one 8 aspects (N, NE, E, SE, S, SW, W and NW) with two different slope angles (20° and 40°) for each aspect. However, users must be aware that a significant small-scale spatial variability of snow cover remain unresolved by this approach (e.g. Weber et al. (2020)).

In addition to simulations provided on the massif geometry, simulations are also performed over a set of 665 observation stations. These sites have been selected to be higher than 600 m a.s.l and to provide snow depth measurements. The selected sites cover the three domains of the reanalysis with 435 observations sites in the Alps, 208 in the Pyrenees and 22 in Corsica. Each site is characterized by its altitude, slope and aspect. A shapefile containing all the stations informations is provided with
110 the dataset. These simulations result from an interpolation between the nearest elevation bands of the corresponding massif and a projection of the direct solar radiation according to the slope and aspect of the station, the time during the day and the information about solar masks from surrounding topography. The main purpose of these local simulations is to compare snow depth simulations to observations in order to assess the performance of the model (see section 4.4) with as few artifacts as pos-

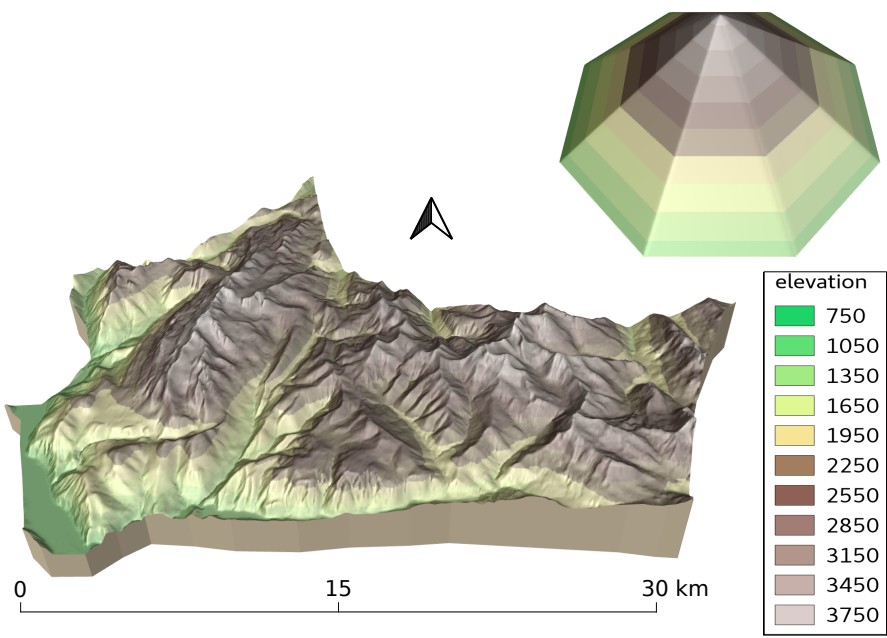

**Figure 3.** Illustration of the ideal representation of a real mountain area (massif des Grandes Rousses, lower left) as topographic classes (upper right) in the S2M reanalysis.

sible due to topographical discrepancies between observations and the simulation configuration. However, it must be kept in mind that the spatial scale of SAFRAN analyses (e.g. precipitation) remains at the massif scale even in this local configuration.

## 2.2 Input data

The S2M reanalysis is only fed by input data to the SAFRAN atmospheric analysis system as described in section 2.4.

### 2.2.1 Meteorological guess used by SAFRAN

The S2M reanalysis uses two different NWP model output as a guess for the SAFRAN analysis :

- The ERA-40 reanalysis (Uppala et al., 2005) between 1958 and 2002, which is based on a uniform data assimilation system (but variable in-situ and satellite network density) over the whole period.

- Operational forecasts of the French global NWP model ARPEGE from 2002 to 2021, which evolved over this time period with on average one major evolution per year.

The use of the ERA-40 reanalysis instead of the most recent ERA-interim or ERA-5 is inherited from previous work (Durand et al., 2009a, b). It is planned to use ERA-5 for future updates of this reanalysis. Outputs from the above-mentioned NWP

| Variables | levels |
|---|---|
| Geopotential (m) | Surface, 1000 hPa, 950 hPa, 900 hPa, 850 hPa, 700 hPa, 500 hPa, 300 hPa* |
| Temperature (K) | 2 m, 20 m, 50 m, 100 m, 250 m, 1000 m, 1500 m, 1000 hPa, 950 hPa, 900 hPa, 850 hPa, 700 hPa, 500 hPa, 300 hPa* |
| Meridian wind ($\text{m s}^{-1}$) | 10 m, 20 m, 50 m, 100 m, 250 m, 1000 m, 1500 m, 1000 hPa, 950 hPa, 900 hPa, 850 hPa, 700 hPa, 500 hPa, 300 hPa* |
| Zonal wind ($\text{m s}^{-1}$) | 10 m, 20 m, 50 m, 100 m, 250 m, 1000 m, 1500 m, 1000 hPa, 950 hPa, 900 hPa, 850 hPa, 700 hPa, 500 hPa, 300 hPa* |
| Relative humidity (%) | 2 m, 20 m, 50 m, 100 m, 250 m, 1000 m, 1500 m, 1000 hPa, 950 hPa, 900 hPa, 850 hPa, 700 hPa, 500 hPa, 300 hPa* |
| Pressure (hPa) | Surface, 20 m, 50 m, 100 m, 250 m, 1000 m, 1500 m |

**Table 1.** List of variables and levels used as meteorological guess. All levels are not always available depending on the period and the grid point (the 300 hPa level is only available in the ERA-40 reanalysis)

systems are used as a preliminary guess at a 6-hour time resolution of the main variables driving the evolution of the snow cover (see Table 1). This guess contains informations both at the surface and at different heights above the surface.

The guess for precipitation since 1958 to 1 August 2017 is obtained using the AURELHY analysis method (Bénichou and Le Breton, 1987) providing a 24-hour climatological accumulation for each massif, depending on the weather type (without any use of the precipitation fields from the NWP model). Since 1 August 2017, 24-hour cumulated ARPEGE precipitation fields are used as precipitation guess and the chronology of precipitation is taken from 6-hour ARPEGE precipitation. This change

is due to the improvements in simulated precipitation and their availability at a 6-hour time resolution, which increases the consistency of the analysis and meaningfulness of the computation of the phase of precipitation. However, it is not possible to extend this approach back in time, using the current NWP model output, since 6-hour resolution precipitation data in ARPEGE is only available since 2017.

**2.2.2 Surface observations**

The most influential observations analysed by SAFRAN are surface observations from various networks. This includes manual observations from the Météo-France climatological network and the dedicated snow observation network ("réseau nivo-météorologique" in French) resulting from a collaboration between Météo-France and mountain stakeholders (in particular Domaines Skiables de France, Association Nationale des Maires de Stations de Montagne, Association Nationale des Di-

145 recteurs de Pistes et de la Sécurité de Stations de Sports d'Hiver). The latter network has been progressively implemented since the 1970s, with observations relevant to the mountain snow cover. These observations are a key part of the analysis system since they often are the only available information for a given area. They include a large range of meteorological and snow cover variables including past weather conditions, information on the rain-snow elevation or 24h height of new snow one or two times per day, which are used to ensure a consistent analysis and check automatic observations. Observations for Andorra

and the Spanish side of the Pyrenees are provided by means of international collaborations for the exchange of snow and meteorological observations (NIVOMET). Automatic observations result from various relevant automated networks, including

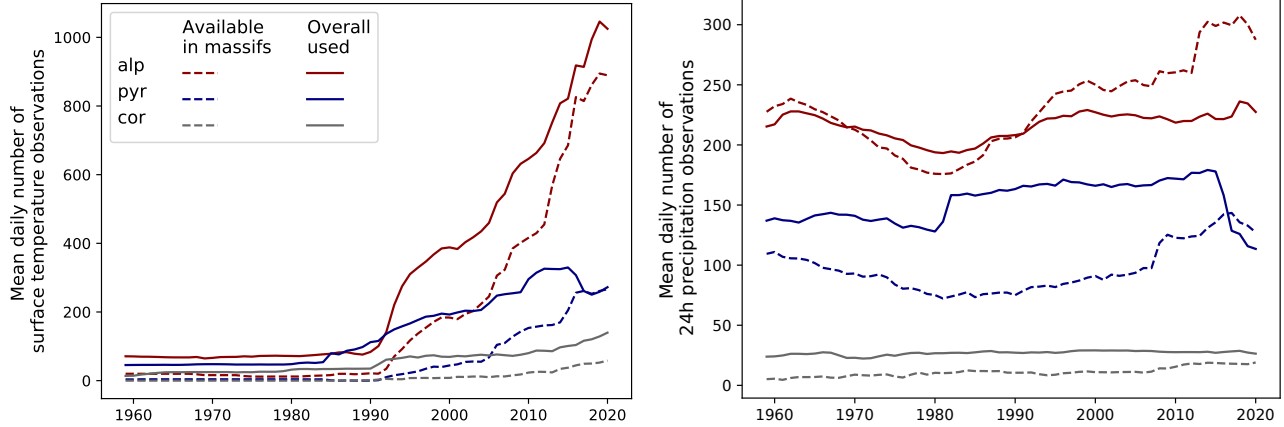

**Figure 4.** Temporal evolution of the daily mean number of surface temperature (at 2 m) and 24-hour precipitation observations available within the massifs limits (dashes) and effectively assimilated (solid lines) for each mountainous area over the period covered by the reanalysis.

the dedicated high elevation Nivose network. All surface observations are characterized according to their representativeness of the surrounding massifs in order to include them in the analysis of different massifs if not enough observations are available within a given massif.

The temporal evolution of the number of daily 2 m surface temperature and 24-hour precipitation observations available in the massifs and effectively used in the assimilation process is shown in Figure 4. Available observations (dashes) only refer to stations within the boundaries of the massifs while the analysis model can use observations coming from more distant low elevation areas (see Figure 2 for the extraction domains of observations). This difference explains that the daily average number

of assimilated observations can be higher than the number of available observations within the simulation domain.
Figure 4 shows a generally growing number of available observations in the mountains over most of the period 1958-2020 in the three areas. This explains the increasing number of assimilated observations and suggests that the share of mountain observations in the reanalysis also rose. The stabilisation (for 2 m-temperature) or decrease (for precipitation) of the number of observations for the five last years of the period is due to a cost reduction in some manual measurements and automatic net-

165 works. The rapid increase of 2 m-temperature observations (Figure 4) starting from the beginning of the 1990s can be explained by the development of automatic networks providing hourly observations in mountain areas, thus adding many more observations than daily precipitation observations. It is an important source of temporal heterogeneity in the dataset (see section 5.2). These figures also show that the three areas have different observation network densities, in particular there are significantly more observations in the French Alps than in the Pyrenees although these two areas have a similar number of massifs.

Figure 5 shows the mean daily number of observations over the 1958-2020 period for the same variables by elevation bands up to 3300 m a.s.l. for the three domains. It highlights the large share of low elevation observations (up to 600 m a.s.l) in the

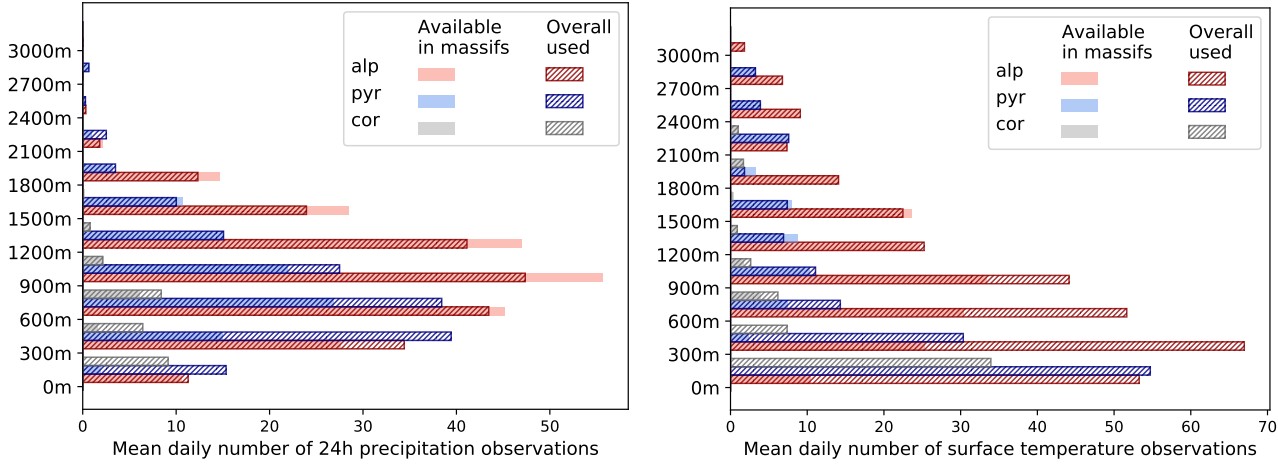

**Figure 5.** Mean daily number of 2 m-temperature and 24-hour precipitation observation sites available (plain colours) and used (hatches) to produce the S2M reanalysis in every 300 m elevation band for the French Alps, Pyrenees and Corsica on average for the 1958-2020 period.

SAFRAN analysis even though these are not necessarily the most representative of high elevation climate. It especially shows that there are very few available observations in the massifs above 3000 m a.s.l. (and even above 2100 m a.s.l for precipitation because the Nivose network does not include precipitation measurements). This implies that most of the specific information relevant to the mountain environment is located within the middle elevation ranges (from 600 m to 2100 m a.s.l.). These elevations typically match ski resort elevations where manual observations are performed in wintertime and most automatic stations are located. The observation network at these elevations in the French Alps is dense enough for the assimilation algorithm to reject some apparently spurious observations whereas at higher elevations and in the two other domains most available observations are usually used in the analysis system. At low elevations (bellow 660 m) many observations from outside the massifs limits are used, including observations from distant flat areas.

For this new version of the S2M reanalysis, a more complete set of observations than the previous one (Durand et al., 2009a) was used, using all observations available in the Météo-France database.

## 2.3 Other observations

SAFRAN also uses two additional sources of information:

- Radiosondes provide vertical profiles used to correct the upper part of the profiles coming from the guess. However very few radiosondes observations are available daily and they are often launched from distant areas.

- Satellite observations are used since 1991 to analyse the cloudiness based on 1 km observed cloud structures which can especially help detect total cloudiness and clouds in valleys.

## 2.4 Short description of SAFRAN

SAFRAN (Durand et al., 1993) is an atmospheric analysis system, which provides the main meteorological variables necessary to drive a land surface model (and in particular a snow cover model) at an hourly time step. Each SAFRAN analysis covers a 24-hour period from 6:00 on day D-1 to 6:00 on day D. It combines the gridded meteorological guess from ERA40 or ARPEGE NWP models (see section 2.2.1) providing vertical profiles of air temperature, humidity, wind speed and direction every 6 hours. An estimation of the 24-hour cumulative precipitation over the analysis period is obtained either from a climatology or directly from the NWP simulated precipitation (see section 2.2.1).

The first step of the SAFRAN analysis of all variables except precipitation is to compute a meteorological guess on its massif geometry (see section 2.1) every 6 hours using the vertical profiles from the NWP model. That guess is corrected by the assimilation of a first set of surface observations (of 2 m-temperature, 2 m-humidity and 10 m-wind) available every 6 hours using an optimal interpolation. The weight of each assimilated observation is based on the horizontal distance between the observation and analysis points. These 6-hour values are then interpolated at an hourly time step. The daily evolution of 2 m-temperature is particularly important and the implemented scheme is based on Martin and Mainguy (1988). A first estimate of the maximum daily 2 m-temperature is performed using the analysis at 12:00. Then the 2 m-temperature profiles are temporally interpolated using a diurnal adjustment depending on the other meteorological variables. If hourly observations are available, a variational assimilation of these observations produces the final hourly simulation for the main relevant atmospheric variables affecting surface processes (i.e., 2 m air temperature, 10 m wind speed, 2 m air humidity, cloudiness, long-wave incoming radiation, and direct and scattered solar radiation).

The precipitation analysis consists in an optimal interpolation between the guess and the different available observations of 24 h cumulative precipitation within the massif followed by a temporal distribution of hourly precipitation according to the NWP-model chronology when available (see section 2.2.1) or simulated hourly relative humidity. The phase of hourly precipitation is determined depending on the simulated $0°C$ isotherm elevation for each aspect and potential past weather condition observations as well as an estimation of the daily fraction of solid precipitation mainly based on the manual observation network ("réseau nivo-météorologique").

Meteorological variables for each elevation of each massif are analysed using a maximum number of :

- 12 observations for 2 m-temperature and 10 m-wind,

- 8 observations for 2 m-humidity,

- 16 observations for precipitation, per massif.

Some final adjustments are made at the end of the analysis, to ensure the physical consistency, such as an adjustment of the elevation of the rain/snow limit with respect to 2 m-temperature.

## 2.5 Short description of Crocus

The SURFEX/ISBA-Crocus (hereafter Crocus) model represents the snowpack as a stratified medium depicted by a dynamical number of numerical layers up to 50. The prognostic variables for each layer are snow mass, density, enthalpy (i.e. temperature and liquid water content), age, and complementary variables for snow microstructure (specific surface area and sphericity). Evolution equations rely on the solving of the diffusion heat equation in this stratified medium with Neumann boundary condition. Phase changes (melting and refreezing) are computed assuming a decoupling with heat diffusion at the model internal time step (900 seconds). Empirical parametrisations are implemented to compute the surface energy fluxes (parametrisations of albedo for solar radiation absorption and parametrisations of sensible and latent heat turbulent fluxes). Other parametrisations allow simulating the main other physical processes: metamorphism, compaction and liquid water percolation. The model was initially developed and described by Brun et al. (1989, 1992). The most up-to-date description of the model was published by Vionnet et al. (2012). A multiphysical version was developed by Lafaysse et al. (2017) to quantify uncertainties associated with the empirical parametrisations. The physical options of Crocus used in the S2M dataset correspond to the default configuration as defined in Lafaysse et al. (2017) except for turbulent fluxes (RI2 option).

## 2.6 Short description of MEPRA

MEPRA is an expert model designed to estimate the avalanche hazard from the snowpack stratigraphy simulated by Crocus, from mechanical diagnosis and expert rules (Giraud et al., 2002). It has been fully implemented in the SURFEX platform, and its outputs are computed and made available with the other diagnostic variables. The general concept in MEPRA is to compare the shear strength to the shear stress in each snow layer. The shear strength is parametrised as a function of density and microstructure variables. For natural release, only the weight of overlying layers is taken into account in the shear stress. An additional load is added in order to compute accidental triggering. Expert rules are defined to compute a hazard index from these mechanical stability indicators, both for natural release and accidental triggering. The system is only applied on 40 degrees slopes, at a internal time step of 3 hours.

## 3 Description of the S2M dataset

### 3.1 S2M dataset description

The S2M reanalysis spans the time period from 1st August 1958 at 6:00 UTC until 1st August 2021 at 6:00 UTC. The atmospheric (FORCING.nc) and snow cover (PRO.nc) datasets are each stored in 63 annual NetCDF files for each mountainous area (French Alps, Pyrenees and Corsica). Two kinds of simulation are available: one on flat terrain only, and one taking slopes into account by projecting the incoming radiation variables according to the slope value and its aspect.

#### 3.1.1 Metadata

The S2M dataset consists of two shapefiles containing all informations concerning the geometry of the simulation (for both massifs and stations geometries) and NetCDF files containing the simulated data. Simulated variables have two main dimensions : a temporal one ("time", the number of days from the previous 1st of August) and the total number of simulation points ("Number_of_points"). Geometry variables that can be used to retrieve specific sub-data from the full yearly files are summarised in Table 2. For example to get snow depth simulation over south heading slope, 40° steep in massif number 1 at 2400 m a.s.l elevation, select the following characteristics for variable "DSN_T_ISBA" :

– massif_num = 1

– slope = 40

– ZS = 2400

– aspect = 180.

A practical code in python is provided to get the simulated snow depth evolution over south heading slope, 40° steep in massif number 1 at 2400 m a.s.l elevation, as well as for one specific date. It also show how to plot the Alps massifs from the corresponding shapefile.

#### 3.1.2 Meteorological variables

Meteorological fields are provided at an hourly time step. The list of SAFRAN output variables is summarised in Table 3. The 0°C isotherm elevation and the rain snow limit are are computed using the whole vertical profile simulated by SAFRAN for each massif.

#### 3.1.3 Snow cover and soil variables

Files containing snow cover and soil variables include data at a daily time resolution (with state variables provided at 6:00 UTC). The list of the output snow cover and soil diagnostics is given in Table 4. Note that some diagnostics have a high diurnal

| Geometry variable | Name | Unit | Comment |
|---|---|---|---|
| Massif Number | massif_num | | Massif number (see Figure 2 or shapefile), only for the massif geometry |
| Station | station | | WMO station number (see shapefile), only for the station geometry |
| Longitude | longitude | degrees GCS | Longitude of the station (see shapefile), only for the station geometry |
| Latitude | latitude | degrees GCS | Latitude of the station (see shapefile), only for the station geometry |
| Elevation | ZS | m a.s.l | Station elevation or every 300 m a.s.l for the massif geometry |
| Slope | slope | degrees | Slope angle of the station or $0°$ (flat) or $20°$ and $40°$ (allslopes) |
| Aspect | aspect | degrees from North | Aspect of the station or every 45 degree for the massif geometry |
| Time | time | days or hours | Since the previous 1st August at 6:00 UTC |

**Table 2.** Description of the metadata of the S2M dataset.

| Variable | Name | Unit | Height |
|---|---|---|---|
| Surface pressure | PSurf | Pa | surface |
| Surface temperature at 2 m | Tair | K | 1.5 m |
| Wind speed | Wind | $\mathrm{m\,s^{-1}}$ | 5 m |
| Wind direction | Wind_DIR | degrees from N | 5 m |
| Specific humidity | Qair | $\mathrm{kg\,kg^{-1}}$ | 1.5 m |
| Relative humidity | HUMREL | % | 1.5 m |
| Rainfall rate over the last hour | Rainf | $\mathrm{kg\,m^{-2}\,s^{-1}}$ | surface |
| Snowfall rate over the last hour | Snowf | $\mathrm{kg\,m^{-2}\,s^{-1}}$ | surface |
| Surface incident long-wave radiation | LWdown | $\mathrm{W\,m^{-2}}$ | surface |
| Direct short wave radiation | DIR_SWdown | $\mathrm{W\,m^{-2}}$ | surface |
| Diffuse short-wave radiation | SCA_SWdown | $\mathrm{W\,m^{-2}}$ | surface |
| Cloudiness | NEB | cloud area fraction between 0 and 1 | |
| $0°C$ isotherm elevation | isoZeroAltitude | m | |
| Rain-snow limit altitude | rainSnowLimit | m | |

**Table 3.** List of SAFRAN output variables, at an hourly time step

variability (snow surface temperature, stability indices,...). A higher temporal resolution may be provided in a future version of this dataset if we receive expressions of interest. The detailed vertical profiles of the prognostic variables of the snowpack are not provided because it would represent a too big data volume at this spatial extent. Tables 5 and 6 provide a list of the variables relevant to the surface energy balance (with or without snow on the ground). Mechanical variables computed by the MEPRA model are given in table 7 and provide estimates of the stability of the snowpack.

| Variable | Name | Unit | Height |
|---|---|---|---|
| Total snow depth | DSN_T_ISBA | m | |
| Snow surface temperature | TS_ISBA | K | surface |
| Total snow reservoir (Snow Water Equivalent, SWE) | WSN_T_ISBA | $\mathrm{kg\,m^{-2}}$ | |
| 1 day new snow thickness | SD_1DY_ISBA | m | |
| 3 days new snow thickness | SD_3DY_ISBA | m | |
| 5 days new snow thickness | SD_5DY_ISBA | m | |
| 7 days new snow thickness | SD_7DY_ISBA | m | |
| 1 day new SWE | SWE_1DY_ISBA | $\mathrm{kg\,m^{-2}}$ | |
| 3 days new SWE | SWE_3DY_ISBA | $\mathrm{kg\,m^{-2}}$ | |
| 5 days new SWE | SWE_5DY_ISBA | $\mathrm{kg\,m^{-2}}$ | |
| 7 days new SWE | SWE_7DY_ISBA | $\mathrm{kg\,m^{-2}}$ | |
| Penetration depth of ram resistance sensor $<$2daN | RAMSOND_ISBA | m | |
| Wet snow thickness | WET_TH_ISBA | m | surface |
| Refrozen snow thickness | REFRZTH_ISBA | m | surface |

**Table 4.** List of instantaneous snowpack variables (at 6:00 UTC) included in the S2M reanalysis

| Variables | Name | Unit |
|---|---|---|
| Net radiation over tile nature | RN_ISBA | $\mathrm{W\,m^{-2}}$ |
| Ground flux over tile nature | GFLUX_ISBA | $\mathrm{W\,m^{-2}}$ |
| Surface albedo | TALB_ISBA | - |
| Downward long-wave radiation | LWD_ISBA | $\mathrm{W\,m^{-2}}$ |
| Upward long-wave radiation | LWU_ISBA | $\mathrm{W\,m^{-2}}$ |
| Downward short-wave radiation | SWD_ISBA | $\mathrm{W\,m^{-2}}$ |
| Upward short wave radiation | SWU_ISBA | $\mathrm{W\,m^{-2}}$ |
| Upward sensible heat flux | H_ISBA | $\mathrm{W\,m^{-2}}$ |
| Upward latent heat flux | LE_ISBA | $\mathrm{W\,m^{-2}}$ |
| Evaporation flux | EVAP_ISBA | $\mathrm{W\,m^{-2}}$ |
| Snow melting rate | SNOMLT_ISBA | $\mathrm{kg\,m^{-2}\,s^{-1}}$ |
| Cumulative rainfall flux | RAINF_ISBA | $\mathrm{kg\,m^{-2}\,s^{-1}}$ |

**Table 5.** List of variables integrated over the past 24 hours at 6:00 UTC

| Variable | Name | Unit | Depth |
|---|---|---|---|
| Soil temperature | TG1 and TG4 | K | 0.5 cm and 8 cm |
| Liquid water content | WG1 | $\mathrm{m^3\,m^{-3}}$ | 0.5 cm |
| Solid water content | WGI1 | $\mathrm{m^3\,m^{-3}}$ | 0.5 cm |

**Table 6.** List of soil parameters simulated by SURFEX

| Variable | Name | Unit |
|---|---|---|
| Depth of high instability layer | DEP_HIG | m |
| Depth of moderate instability layer | DEP_MOD | m |
| Accidental risk index | ACC_LEV | - |
| Natural risk index | NAT_LEV | - |
| Type of avalanche | AVA_TYP | - |

**Table 7.** List of snowpack stability indices simulated by MEPRA

## 4 Results

### 4.1 Impact of the change of precipitation guess

The impact of the temporal heterogeneity introduced by the different precipitation guess has been evaluated over one single season (2017-2018) by comparing a simulation made with precipitation guess based on the AURELHY analysis method to the reference one (with precipitation guess from ARPEGE). The comparison of the simulated snow depth mean deviation and root mean square deviations (RMSD) for these 2 configurations didn't show any significant impact on the performance of the system. Furthermore, the simulated annual precipitation amount with the precipitation guess based on the AURELHY analysis method seems to be slightly higher (about 3% on average) than those simulated with precipitation guess from ARPEGE : for the season 2017-2018 the average total precipitation over the 665 stations of the simulation with the guess based on the AURELHY analysis method and from ARPEGE respectively is 1507 mm (resp. 1464 mm) with accumulation ranging from 728 mm (resp. 675 mm) up to 3125 mm (resp. 3242 mm).

### 4.2 Meteorological and snow cover climatology and inter-annual variability from the S2M reanalysis

The S2M reanalysis provides a comprehensive appraisal of the climatology and inter-annual variability of meteorological and snow cover conditions in the French Alps, Pyrenees and Corsica. Here we introduce examples on how the data can be exploited, enabling exploration of the dataset across its three main dimensions (time, horizontal dimension - massif - and elevations), using the French Alps as an example and focussing on the five indicators as follows:

- air temperature at 2 m (seasonal and annual mean values)

- total precipitation (seasonal and annual cumulative values)

- fraction of solid precipitation (seasonal and Nov-Apr mean values)

– mean snow depth (seasonal and Nov-Apr mean values)

- snow cover duration (number of days with a non zero simulated snow depth)

Figure 6 shows annual values, aggregated over all the massifs of the French Alps of these five indicators for three different elevations. The envelops represent the variability between the different massifs of the Alps. The amplitude of variations of the annual mean 2 m-temperature (Figure 6a) at a given elevation is lower than 2°C with low variability between the different

massifs (shaded areas around the lines). The mean temperature trends simulated by the S2M reanalysis are +0.10°C per decade at 2700 m, +0.26°C per decade at 1800 m and +0.18°C per decade at 900 m. These trends are significant (all p-values of the trend slope significance are lower than 0.014). The variation of the mean fraction of solid precipitation in winter (Figure 6c) is much larger with an amplitude of about 20 % (except at high elevation where the 2 m-temperature is low enough to have almost only solid precipitation). Annual precipitation (Figure 6b) vary a lot with an inter-massif variability for a given year reaching

about 30 % of the mean. This is consistent with large variations of the mean snow depth in winter (Figure 6d) and the snow cover duration (Figure 6e) from one year to the other as well as between the different massifs.

As shown in Figure 6, averaging the simulated values over the whole Alps hides a strong spatial variability between the massifs. Figure 7 shows that the mean simulated snow depth in winter at 1800 m a.s.l. over the period 1961-1990 ranges from

about 10 cm on the least snowy massif of the Southern Alps up to about 1 m for some of the northern massifs of the Alps. The temporal variability of the S2M snow depth can also be visualised with a time series of miniature maps : Figure 8 shows the yearly anomalies of the average snow depth in winter at 1800 m a.s.l. against the reference 1961-1990 (shown in Figure 7). This visualisation highlights the strong inter-annual variability of the winter snow cover, with fluctuations that can be greater than 100% of the mean value over the reference period. It also points the spatial variability for one given year with anomalies

that can be positive for some massifs and negative for other massifs.

Figure 9 explores the vertical dimension of the dataset, as well as the simulated trends for the different seasons of the year. It compares the mean of the different variables over the Alps for each season at different elevations over the periods 1960-1990 and 1990-2020 (left), as well as the difference between the two periods (right). Only elevations between 900 m a.s.l and 3000 m

a.s.l are considered because the lack of input observations outside this range of elevations (see Figure 5) reduces the quality of the reanalysis.

In terms of mean 2 m-temperature over the Alps, Figure 9 (a, b) shows different patterns depending on the season :

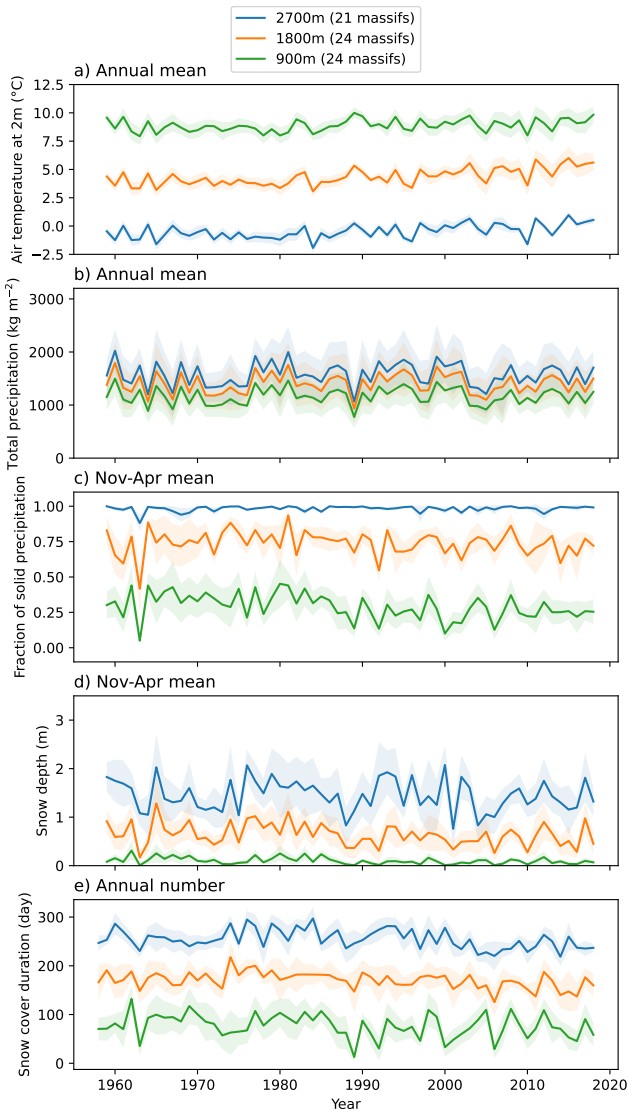

**Figure 6.** Evolution of the annual mean air temperature at 2 m (a), total precipitation (b), winter (November to April) mean of the fraction of solid precipitation (c), total snow depth (d) and snow cover duration (e) aggregated over all the massifs of the French Alps at different elevations. The shadings represents the variability between the massifs.

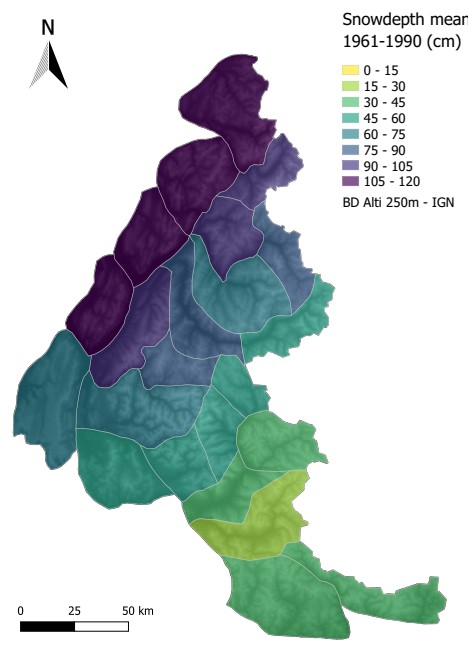

**Figure 7.** Mean snow depth in winter (from November to April) at 1800 m a.s.l. for the different massifs of the French Alps over the period 1961-1990.

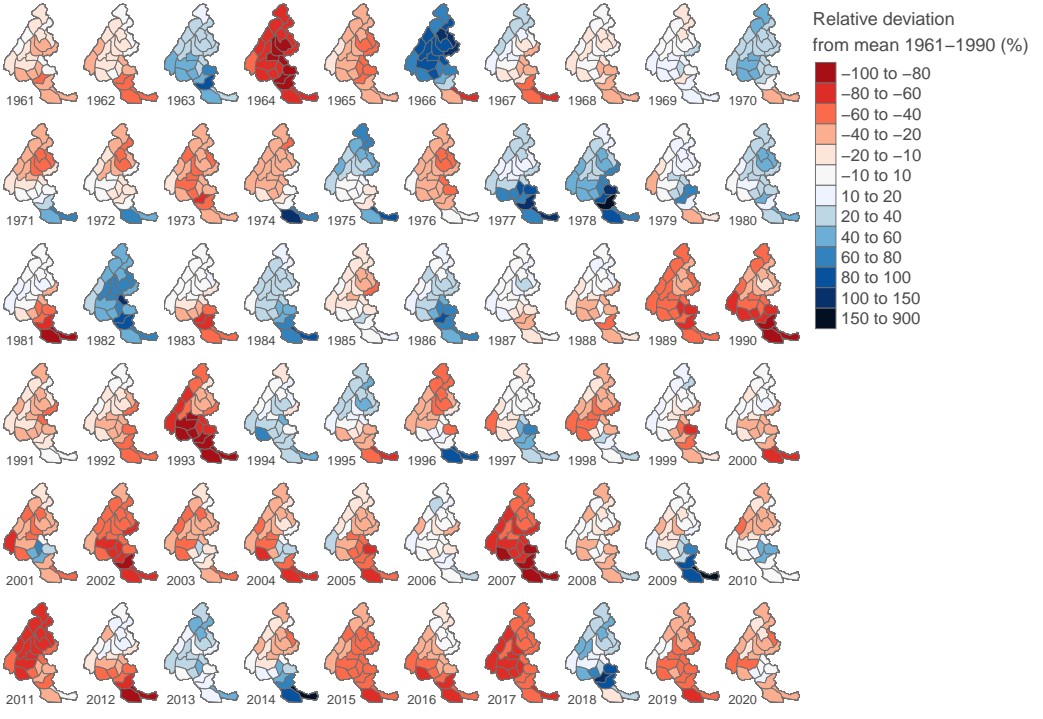

**Figure 8.** Mean deviation of natural snow depth (in meters) from the mean snow depth of the period 1961-1990 in winter (from November to April) at 1800 m a.s.l.

- In spring and summer the S2M reanalysis presents a positive trend of +0.2°C per decade at low elevations up to +0.4°C per decade around 2100 m a.s.l.

- In fall and winter the simulation has a slight negative trend of about -0.1°C per decade.

Beaumet et al. (2021) compared the 2 m-temperature trends of the S2M reanalysis to that of the ones simulated by a model run of the MAR regional climate model driven by the ERA-20C reanalysis and SPAZM reanalysis (Gottardi et al., 2012) and showed that the S2M reanalysis simulates smaller trends both in winter and at high and low elevations in summer. Further analysis of these simulated trends and a comparison to observations is presented on section 4.3.3.

Concerning precipitation, Figures 9 (c, d) shows that the mean of total precipitation in S2M over the Alps in the last three decades is higher in summer (rise of about 3%) and fall (rise of about 10%) and lower in winter and spring (drop of about 3%) than in the three previous decades at all elevations. But the fraction of solid precipitation (Figure9, e and f) decreases in all elevations and seasons, except at high elevations in summer where the total precipitation increases and the temperature is low enough to have frequent snowfalls in summer. Theses trends are consistent with a general decrease of the average total snow depth over the Alps at all elevations and for all seasons between the two considered periods (Figure 9, g and h), except for a small increase at high elevations in autumn. This exception is explained by an increase of the simulated total precipitation (Figure 9, c and d) combined with a simulated cooling of more than 0.1°C per decade (Figure 9, b) resulting in a significant increase of the fraction of solid precipitation (Figure 9, f). The comparatively low amplitude of the increase of the simulated snow depth (only few centimetres, Figure 9, h)) may be explained by an averaging effect and the very short lifespan of snow on ground at this season.

## 4.3 Evaluation of simulated 2 m-temperatures and precipitation

### 4.3.1 Data and Methods

Among the local simulations described in section 2.1, a specific evaluation of the simulated air temperature at 2 m and precipitation was made using homogenised series of monthly observations across the French Alps (mostly located at mid elevations) between 1960 and 2012. These series provide observations of monthly mean of daily maximum (14 stations) and minimum (9 stations) 2 m-temperatures as well as monthly precipitation (43 stations) and have been homogenised using the HOMER software (Mestre et al., 2013). However it is important to consider that the corresponding raw daily observations are assimilated in the S2M reanalysis described in this paper and therefore do not constitute a fully independent evaluation dataset. Consequently, another reanalysis was performed after removing these observations from the assimilation process for an independent evaluation. A third reanalysis was made without the assimilation of any 2 m-temperature observation to identify the impact of the assimilation of these observations on the simulation performance.

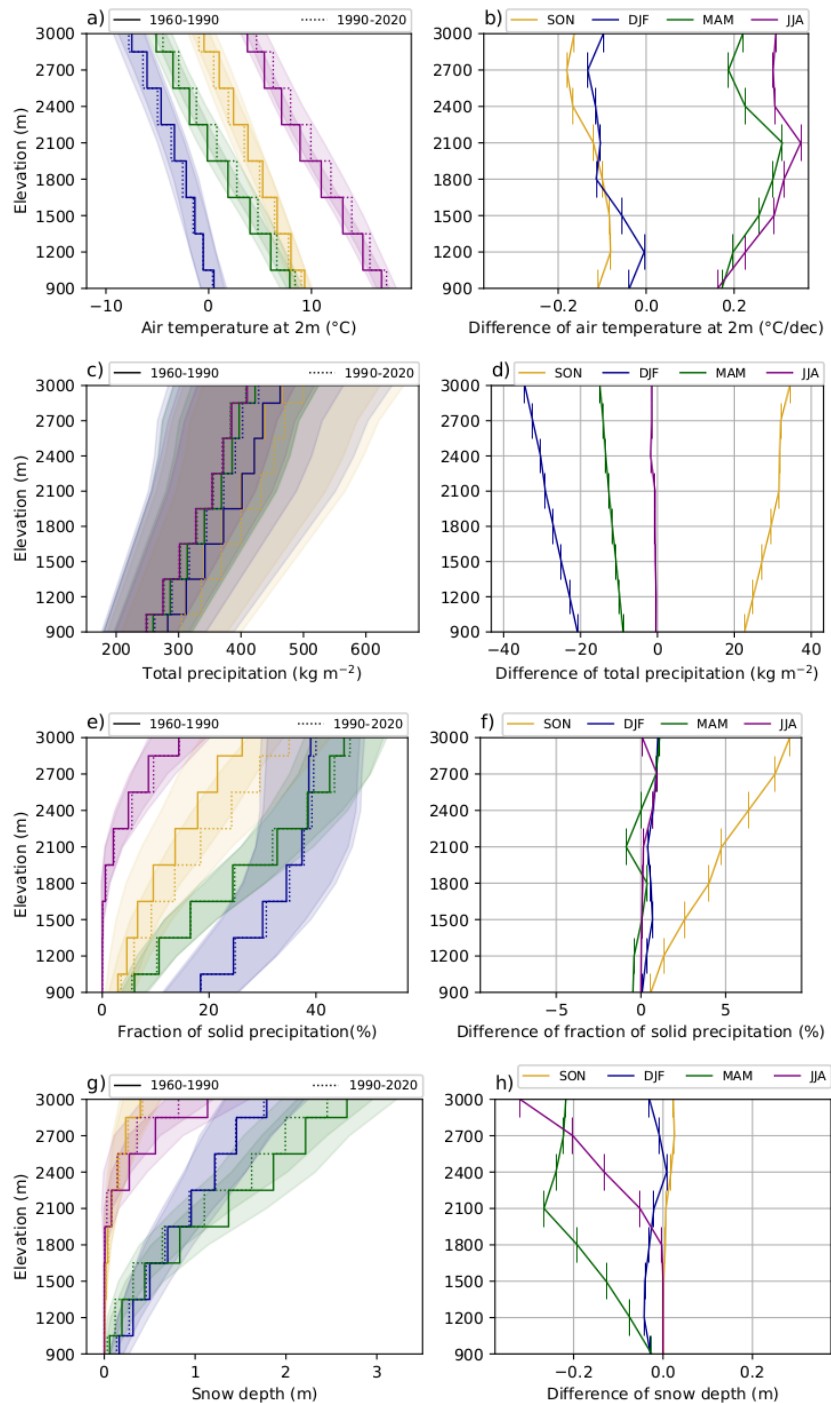

**Figure 9.** Difference of mean simulated 2 m-temperature (a, b), total precipitation (c, d), fraction of solid precipitation (e, f) and snow depth (g, h) for the different elevations and seasons between the periods 1960-1990 and 1990-2020 over the Alps. The envelops represents the variability between the massifs. Vertical bars on the right column indicate the 300 m elevation range that is covered by the corresponding dots.

For the evaluation of 2 m-temperature simulations, it is necessary to take into account the bias due to the hourly resolution of SAFRAN simulations whereas the observed minimum and maximum 2 m-temperatures often stands in-between two hourly 2 m-temperatures. These biases can be estimated by comparing observation series of hourly 2 m-temperatures and hourly minimum and maximum 2 m-temperatures using the raw hourly observations of the evaluation sites (available since the mid 1990s until August 1st 2020). These data show that the minimum value of hourly 2 m-temperatures is on average about 0.2°C higher than the absolute daily minimum and the maximum value of hourly 2 m-temperatures is on average about 0.4°C lower than the absolute daily maximum. The 0.2°C gap between the minimum and maximum 2 m-temperatures shifts highlights that the daily 2 m-temperature evolution is not a perfect sinusoid and presents a day/night asymmetry (Martin and Mainguy, 1988) : a typical daily 2 m-temperature evolution curve for a clear sky day exhibits a narrow maximum and flat minimum. Thus, the skill of the mean simulated 2 m-temperature can not be strictly assessed from the skill of the average of minimum and maximum 2 m-temperatures. The available evaluation dataset does not allow to provide an equivalent evaluation of hourly 2 m-temperatures or even daily mean 2 m-temperatures. Thus the 2 m-temperature evaluation consists of direct comparison between (hourly) simulated and (absolute) observed minimum and maximum 2 m-temperatures. This implies that a simulation perfectly matching the corresponding observations is expected to display a mean deviation of +0.2°C on minimum 2 m-temperature and a mean deviation of -0.4°C on maximum 2 m-temperature.

For the two climatological periods and the three simulations, the mean deviation and the root mean square deviation (RMSD) between monthly simulated and observed values were computed without taking into account the season of the year.

Evaluation data were also used in section 4.3.3 to assess and discuss the relevance of simulated trends of 2 m-temperature and precipitation in winter and summer presented in section 4.2 and based on the evaluation informations of section 4.3.2.

### 4.3.2 Evaluation of minimum and maximum 2 m-temperatures and precipitation

The box plots in Figure 10 show the variability of the scores among the different sites within each elevation range and the notches indicate the confidence interval of the median obtained by a bootstrap sampling of the considered stations.

Figure 10 (a and b) presents the RMSD between the simulated and observed minimum and maximum 2 m-temperatures. For both variables, the RMSD significantly decreases in the later period when more observations are assimilated (red and blue boxplots) and remains constant or even increases when no observations are assimilated (grey boxplots). Similarly, the absolute values of the mean deviation between the maximum and minimum of hourly simulated 2 m-temperatures and the observed daily maximum and minimum 2 m-temperatures (Figure 10, c and d) both decrease over time only for simulations with 2 m-temperature observations assimilation. These figures clearly show that the assimilation of 2 m-temperature observations have a major impact on the simulation quality and that the increasing number of these observations over time results in an improve-

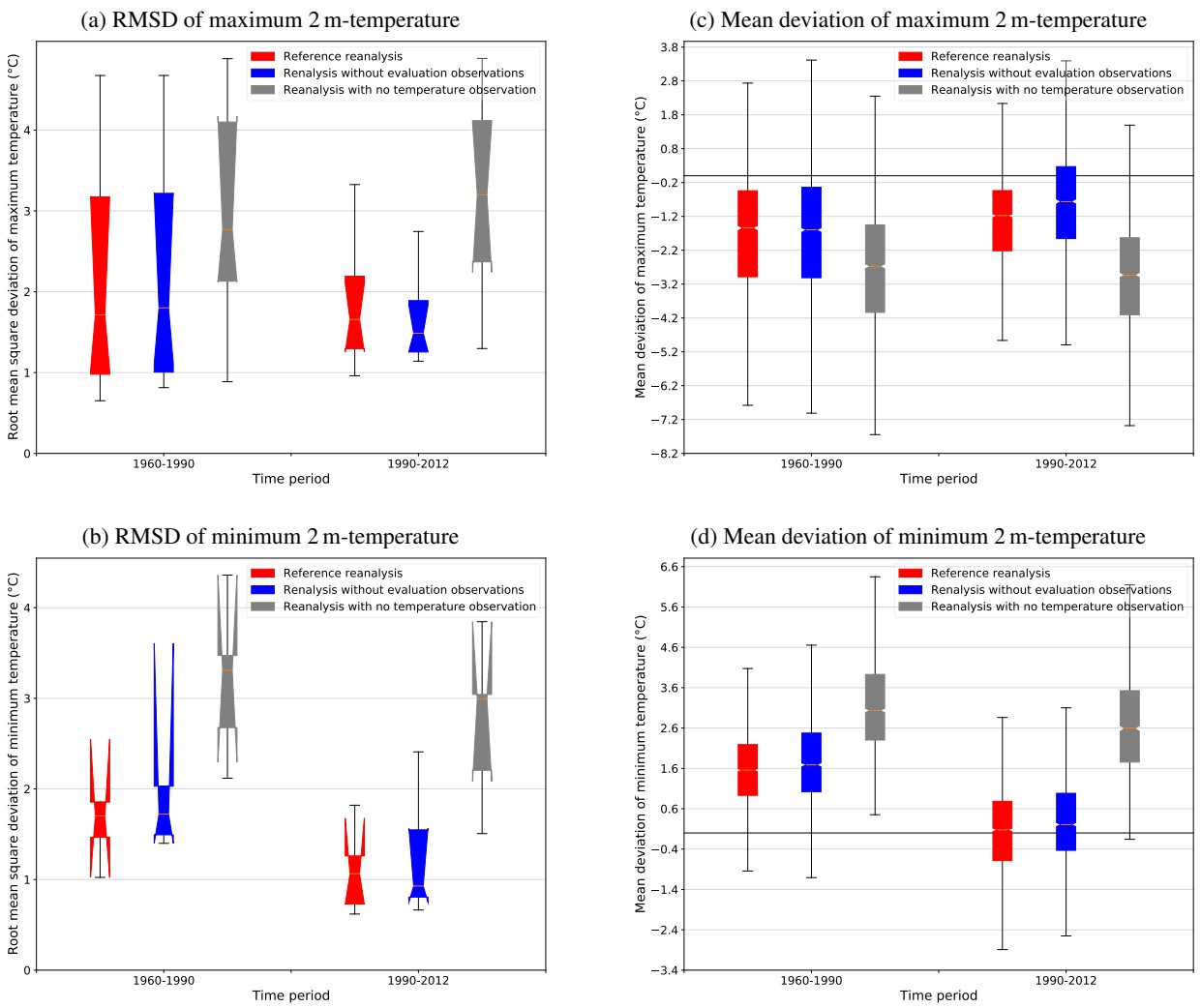

**Figure 10.** Root mean square deviation (left) and mean deviation (right) between the simulated monthly mean of maximum (top) and minimum (bottom) daily 2 m-temperatures and the corresponding homogenised series over the periods 1960-1990 and 1990-2012 for three different version of the S2M reanalysis (see section 4.3.1).

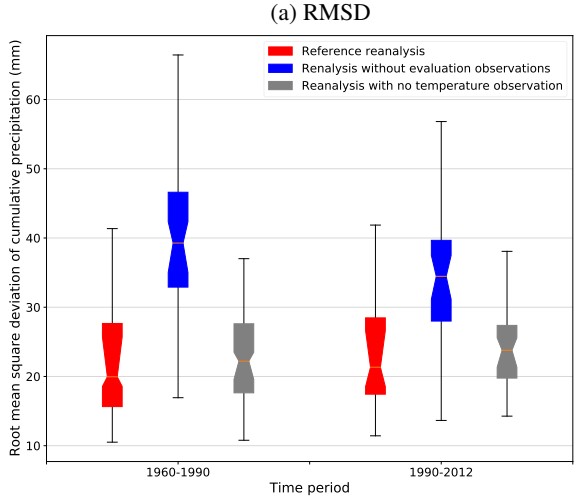
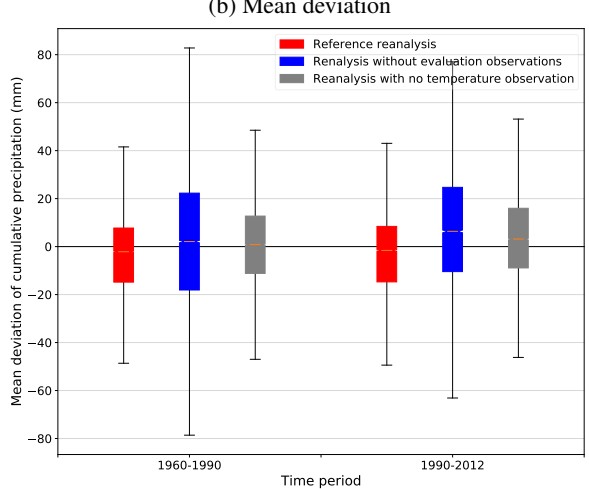

**Figure 11.** Root mean square deviation (left) and mean deviation (right) between the simulated monthly cumulated precipitation and the corresponding homogenized series form the periods 1960-1990 and 1990-2012 for three different version of the S2M reanalysis (see section 4.3.1).

ment of the simulation.

The magnitude of the mean deviation evolution in Figure 10 also provides important informations. The mean deviation of the minimum 2 m-temperature drops between the two periods by about 1.4°C from around 1.6°C to around 0.2°C on average. This 0.2°C residual deviation is expected as it is close to the mean deviation between observed minimum of hourly 2 m-410 temperatures and the daily minimum (see section 4.3.1). On the contrary, the mean deviation of the maximum 2 m-temperature rises by roughly 0.3°C (reference reanalysis) and 0.8°C (independent simulation) on average between the two periods. In this case, a residual bias of about 0.8°C for the reference reanalysis and of about 0.4°C for the independent simulation in the later period remains unexplained compared to the 0.4°C observed difference between hourly maximum and absolute maximum 2 m-temperatures (see section 4.3.1). The main factor of this clear improvement of minimum and maximum 2 m-temperature 415 simulation is the dramatic increased number of assimilated 2 m air temperature observations since the beginning of the 1990s (see Figure 4). This temporal heterogeneity is even more significant since an important part of these new observations are hourly observations. These hourly observations are crucial to accurately simulate the diurnal cycle of 2 m air temperature. Thus the assimilation of more observations over time tends to improve the simulation of the daily variations of 2 m air temperature and to bring it closer to observations for both maximum and minimum temperatures. However, the reduction over time of the 420 warm bias in minimum 2 m-temperature is stronger than the reduction over time of the cold bias in maximum 2 m-temperatures.

The evaluation of precipitation exhibits no strong systematic bias for any simulation or period (Figure 11, b), but a stronger dispersion of the mean deviation for the independent simulation. This is confirmed by the fact that the RMSD (Figure 11, a) is

by far higher for the independent simulation than for the two other simulations despite an improvement over time. As expected, removing all 2 m-temperature observations from the assimilation process does not affect much the precipitation analysis (only minor effects but a major impact exists on the precipitation phase) but removing few precipitation observations (blue boxplots) has a stronger negative impact due to the small number of available precipitation observations (see Figure 4).

### 4.3.3   Trends of minimum and maximum 2 m-temperatures and precipitation

Here, climatological trends are defined by the difference between the mean of a variable over two 30-year long periods (e.g. 1990-2020 and 1960-1990).

Figure 12 compares the simulated and observed differences of minimum (a) and maximum (b) 2 m air temperatures and precipitation (c) between two climatological periods at different elevations in winter and summer. Since the series of observation only cover the period from 1960 to 2012, two different climatological periods are considered for the most recent years (1990-2012 and 1990-2020) for direct comparison.

Figure 12 (a) shows that the order of magnitude of the simulated trends of minimum 2 m air temperature is underestimated by about 0.4°C both in summer and winter, with negative simulated trends instead of positive observed ones. This can be linked with the result of section 4.3.2 that points out that the improvement of the simulation of minimum 2 m-temperature over time generates an artificial cooling of about 1.4°C. On the contrary, Figure 12 (b) shows that simulated trends of maximum 2 m air temperature fit the observed ones much better, especially at low elevations, despite the artificial warming of about 0.3°C highlighted in section 4.3.2.

The magnitude of this warm bias on maximum 2 m-temperature only partially balances the cold bias on minimum 2 m-temperature, resulting in an artificial cooling of the simulation relatively to observations in terms of mean air temperature at 2 m. This pattern is even more pronounced when considering only the winter season (see Figure 16 of Appendix B), which explains the negative 2 m-temperature trend simulated in winter as noticed in section 4.2. Besides, Figure 12 (b) shows that the simulated trends of maximum air temperature at 2 m in summer significantly increases with elevation up to about 1800 m a.s.l. However this elevation dependency of the simulated trend of maximum 2 m-temperature is not visible in station observations.

The dispersion of the observed trends of precipitation in Figure 12 (c) indicates a high spatial variability of these trends. The fact that the simulated trend of the mean precipitation over the Alps stands in the middle of the scatter plot suggests a good agreement between simulated and observed trends of precipitation both in summer (no simulated trend) and winter (slightly negative simulated trend).

### 4.4   Evaluation using snow depth observations

The S2M reanalysis 1958-2021 has been evaluated over the period 1983-2020 by comparing simulated snow depth values to independent observations from the 665 observation sites introduced in section 2.1. The use of total snow depth for the evaluation of the performance of the simulation is motivated by the fact that the analysis system does not use any snow cover

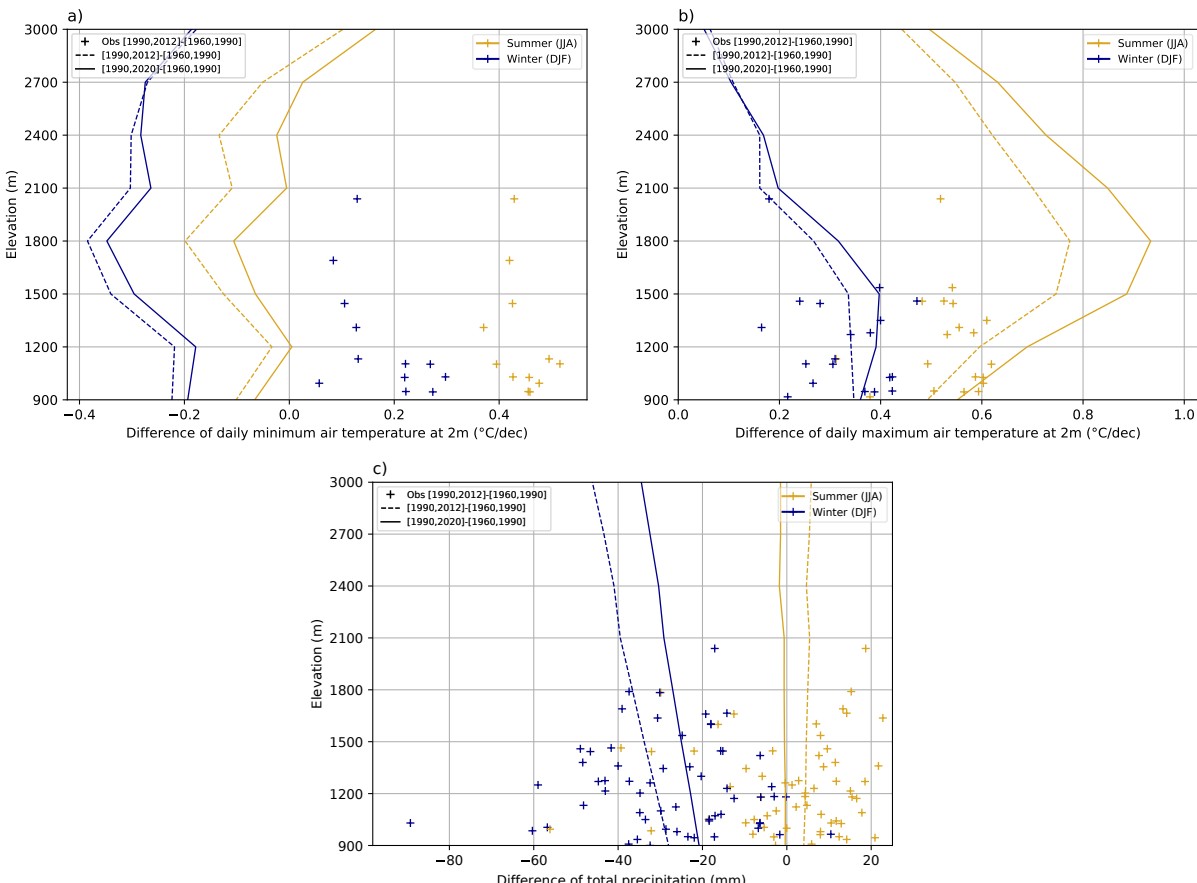

**Figure 12.** Difference of the mean simulated daily minimum (a) and maximum (b) 2 m air temperature and total precipitation (c) for different elevations in summer and winter between the climatological periods 1990-2020 and 1960-1990 (solid line) and 1990-2012 and 1960-1990 (dotted line) over the Alps. Crosses represent the corresponding observed difference on a set of homogenized observation series between the periods 1960-1990 and 1990-2012.

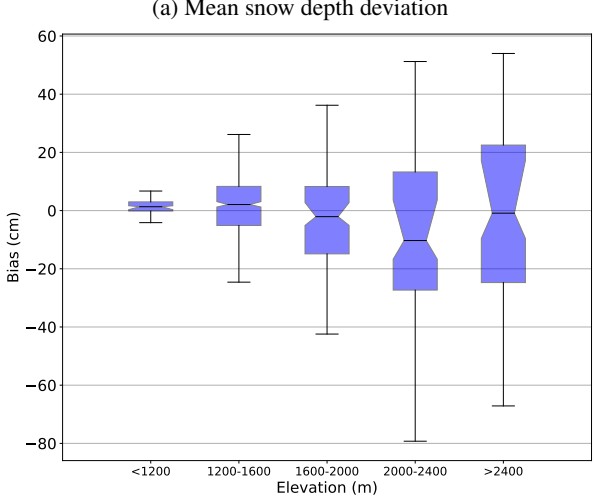

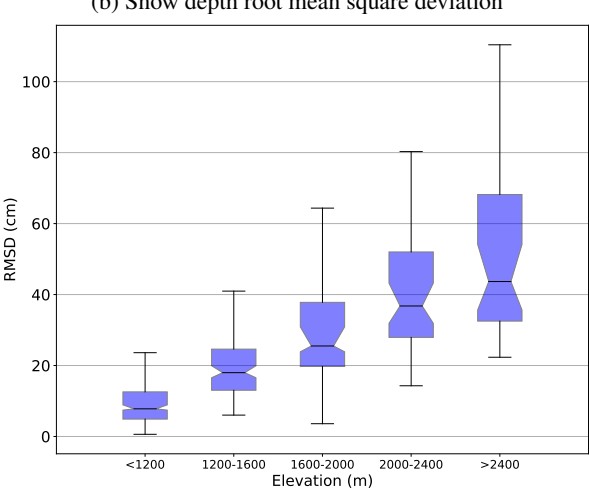

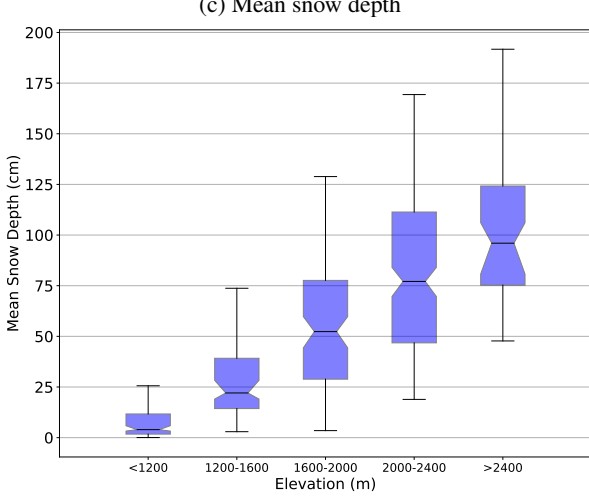

**Figure 13.** Mean deviation, root mean square deviation between the simulated and observed snow depths values and mean simulated snow depth on the 665 validation sites grouped by elevation range.

observation. In addition, snow cover simulations in the S2M reanalysis depend on both the meteorological analysis and the snow cover model. Snow depth is thus an integrated indicator of the overall performance of the full model chain. Last, it is the only variable that is widely available with comparatively low observation errors. The main limitations are its low spatial representativeness, due to the large spatial variability of snow depth at all scales and the temporal coverage of the available observations. Using a large number of snow depth observations partly mitigates the effect of the spatial variability and the evaluation focuses on the last four decades of the reanalysis since very few snow depth observations are available before 1983. We computed two scores for each observation station : mean deviation and RMSD, by taking into account data from October 1st to June 30th of each year.

Figure 13 provides an overview of the ability of the system to simulate snow depth values, with respect to observations, depending on the elevation. This figure compares the simulated and observed snow depth values on the 665 evaluation sites, grouped by elevation range. Figure 13 shows no strong systematic deviation, with a confidence interval of the median deviation always covering both positive and negative values which means that the number of sites with a positive deviation is not significantly different from the number of sites with a negative deviation. The confidence interval is larger at higher elevations due to the lower number of evaluation sites. The sign of the biases is not systematic although slight negative biases tend to prevail above 1600 m a.s.l. RMSD medians increase with elevation from less than 10 cm to around 40 cm in a consistent way with the increase of the mean snow depth but with a different rate. The ratio between RMSD and the mean snow depth is higher at low elevations (almost 100% at elevations below 1200 m a.s.l) and tends to decrease with elevation.

Figures 14 (a) and 14 (b) show the spatial variability of the simulation performance by grouping evaluation sites by French administrative departments (NUTS-3 administrative level in Europe, see Figure 2) in order to have a sufficient number of observations for each unit. There are typically two to six massifs per department.

Figure 14 (a) shows no systematic bias for most departments even if snow depth simulations seems to be slightly overestimated in the northern and central Alps and underestimated in the southern Alps and central and eastern Pyrenees as well as in Corsica. Figure 14 (b) shows that the mean RMSD stands between 15 cm and 30 cm for all departments of the Alps and central and Pyrenees. RMSD is lower in Corsica, with a larger dispersion of the confidence interval of the mean due to the lower number of evaluation sites. Comparing these RMSD to the mean simulated snow depth (Figure 14, c) shows that despite generally higher snow depth values in the Alps, the corresponding simulation errors are not significantly higher over the departments of the Alps than on other departments, indicating a better performance of the simulation chain. This can partly be explained by the fact that there are more available meteorological observations for the SAFRAN analysis in the French Alps than in other massifs (see Figure 4). The evaluation on Corsica does not give much information since the lower number of evaluation sites leads to a huge dispersion of the confidence interval of the RMSD combined to low mean simulated snow depths.

This evaluation shows that the S2M reanalysis is able to simulate a variable that is not assimilated and cumulates errors from both the meteorological analysis and the snow cover model with no systematic bias and moderate deviations to observations.

Another evaluation of the performance of the simulation over time is presented in Figure 15. It compares the mean deviation (a) and RMSD (b) between snow depth simulations of the S2M reanalysis and the simulation with no 2 m-temperature

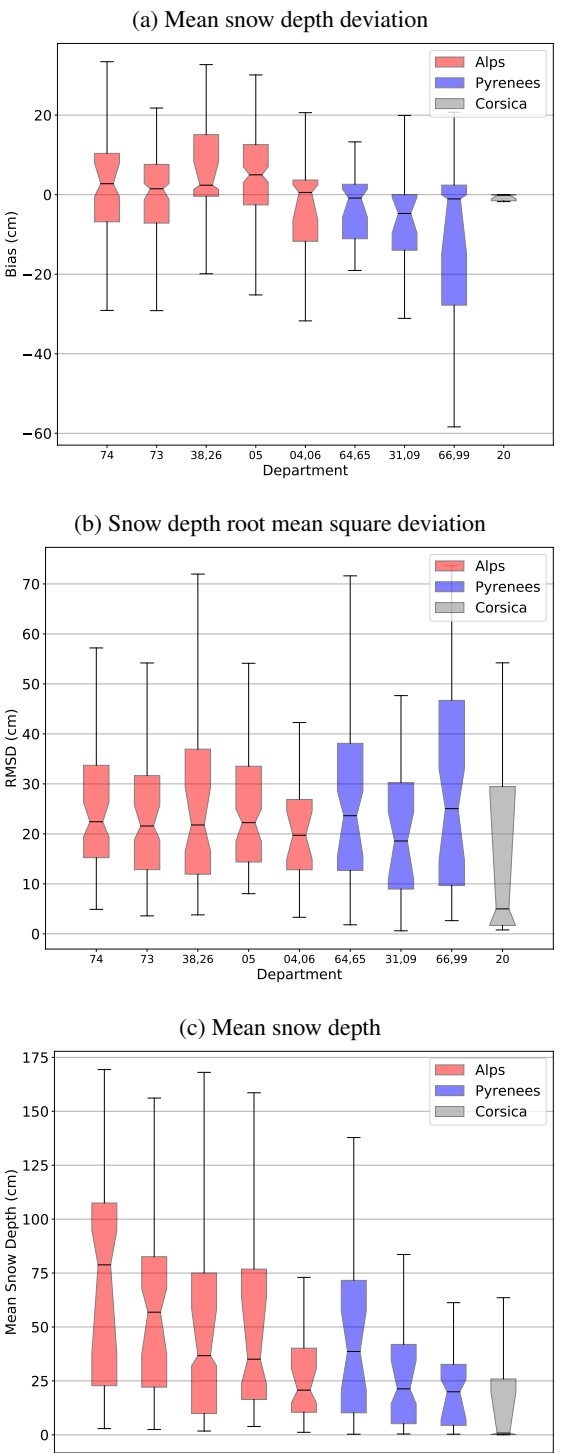

**Figure 14.** Mean deviation and root mean square deviation between the simulated and observed snow depths values and mean simulated snow depth on the 665 evaluation sites grouped by administrative departments. Department numbers 73 and 74 correspond to the Northern Alps, 38, 26 and 05 to central Alps, 04 and 06 to Southern Alps, 64 and 65 cover Western Pyrenees, 31 and 09 central Pyrenees, 66 and 99 Eastern Pyrenees, and 20 Corsica). See Figure 2 and Table 8 for more details.

observations assimilation to their corresponding observations for the last four decades. While the reference S2M reanalysis shows no systematic bias for the four decades, removing 2 m-temperature observations from the SAFRAN analysis introduces a constant negative bias of around 10 cm on average. In addition, the RMSD for the reference reanalysis is always lower by a few centimetres than the simulation with no 2 m-temperature observation (see Figure 15) despite a mean simulated snow depth systematically higher by about 20%. For both simulations, the RMSD tends to decreases over time except for the last decade where the mean simulated snow depth is significantly higher. This consolidates the results of section 4.3.2, confirming that the reanalysis described in this paper provides an optimal simulation at all time by considering all available information at that time. The downside of this is that the simulated trends are not fully representative of the climatological trends due to the temporal heterogeneity of available observations.

## 5   Discussion

In this study, we introduce the open access S2M reanalysis, which provides meteorological and snow cover data for the French Alps, Pyrenees and Corsica for the time period 1958-2021. This dataset enables a large number of research and operational applications, but several limitations need to be considered. Here we review the main strengths and weaknesses of this dataset, and their consequences.

### 5.1   Main assets and known uses

The S2M reanalysis makes it possible to take into account meteorological and snow cover conditions for any time between 1958 and 2021 in the French high elevation mountain regions. It supersedes the SCM reanalysis, which has been developed by Durand et al. (2009a) in the early 2000s. SCM and then S2M have been used to address a wide range of applications in various scientific domains. The S2M dataset was used as a reference for evaluating snow cover simulations driven by the NWP model AROME (Queno et al., 2016; Vionnet et al., 2019). SCM and S2M reanalyses have been used in a number of studies addressing glacier mass balance in the French Alps (Gerbaux et al., 2005; Réveillet et al., 2018; Bolibar et al., 2020; Peyaud et al., 2020) or hydrological simulation of alpine mountainous catchments (Lafaysse et al., 2011). Pellarin et al. (2016) used S2M soil thermal state and the overlying snow cover to investigate the potential of L-band satellite measurements to improve soil moisture retrievals. Although climate trends may be questionable, the S2M dataset provides a robust climatological baseline for mountain regions. It has been used as a reference for adjusting climate change projections with statistical downscaling techniques (Lafaysse et al., 2014; Verfaillie et al., 2017). This methods leads to the provision of meteorological forcing driving files corresponding to future climate time series, on the same geometry and data format as S2M reanalysis forcing files, enabling homogeneous post-processing including running SURFEX/ISBA-Crocus for natural (Verfaillie et al., 2018) and managed (Spandre et al., 2019) snow in ski resorts. There are increasing examples where S2M provides relevant data to investigate the links between vegetation, meteorological and snow conditions (Francon et al., 2020) or extreme events (Corona-Lozada et al., 2019). S2M is also involved in various snow cover process studies (Vionnet et al., 2013; Tuzet et al., 2020) and a foun-

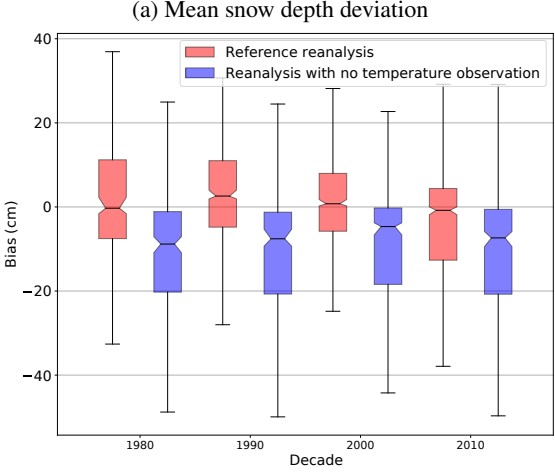

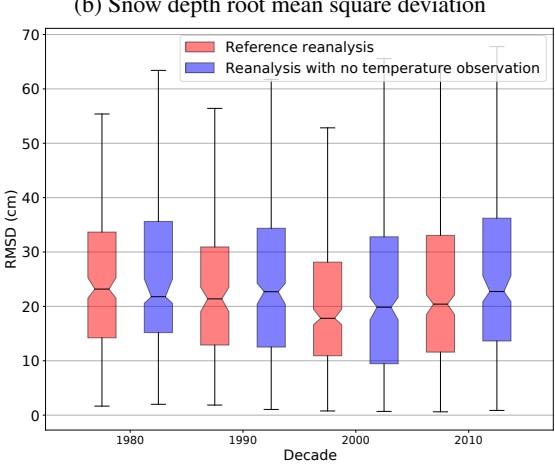

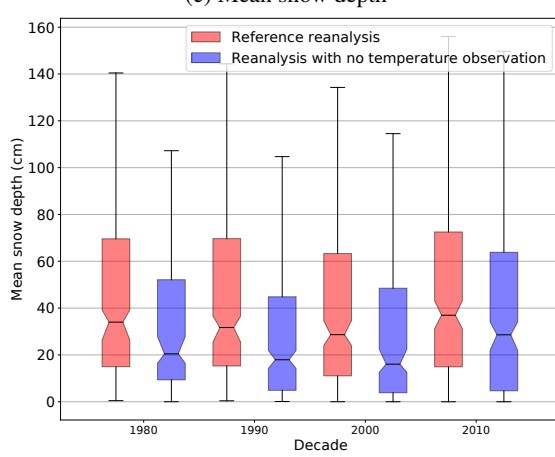

**Figure 15.** Mean deviation (a) and root mean square deviation (b) between the simulated and observed snow depth values and mean simulated snow depth (c) for the the four decades between 1980 and 2020 for the S2M reanalysis and the simulation with no 2 m-temperature observation assimilated (see section 4.3.1).

dation for innovative developments towards the assimilation of remotely sensed and in situ snow cover observations in the
simulations (Viallon-Galinier et al., 2020; Deschamps-Berger et al., 2020; Cluzet et al., 2021).

## 5.2 Temporal heterogeneity

A number of studies rely on S2M for the analysis of meteorological and snow cover trends at climatic scale (Spandre et al., 2019; Lopez-Moreno et al., 2020). For these applications, the main caveat in using S2M lies in the temporal heterogeneity of the dataset, in particular the strong changes in 2 m-temperature observations across the mountain regions considered during the full time period of the reanalysis. The S2M reanalysis for the period 1991-2020 is produced using significantly more observations than the simulation over the period 1961-1990 (Figure 4) and we established in section 4.3.3 that this has a significant impact on the simulated 2 m-temperature trends. This is superimposed on changes in meteorological guess in 2002 (from ERA-40 to ARPEGE) and changes intrinsic to ERA-40 and ARPEGE: assimilation of satellite observations in ERA-40 is known to be responsible for temporal breaks (Sturaro, 2003; Sterl, 2004). Over the time period from 2002 to 2021, the ARPEGE analysis was even more affected by various changes in its physical parametrisation as well as its changes of horizontal resolution over time. Similarly to many other reanalysis systems, the original purpose of the S2M reanalysis was not to provide a system dedicated to the analysis of climate trends, but rather the best available estimate of meteorological and snow cover conditions for every day within the covered time period, hence heterogeneity was allowed in the input data to the S2M reanalysis. This choice is corroborated by the various temporal scores presented in this study showing a clear improvement of the simulation over time correlated to the growing number of available observations. The downside of this temporal refinement is the introduction of artificial biases in various simulated variables that can either compensate or amplify actual climatological trends as presented in this study for 2 m-temperature in winter. Thus, trend analysis using S2M must be considered with high caution, especially for air temperature at 2 m, but also probably for other sensitive variables such as snow depth (see Figure 4 of Verfaillie et al. (2018) at Col de Porte) although very few observations series allow an accurate characterisation of long term trends.

A further evaluation of the simulated snow cover duration values is planned in the coming years, for example products from Hüsler et al. (2014) as evaluation data.

## 5.3 Future updates

The S2M dataset is intended to be updated every year to extend its time coverage period and take into account evolutions of the various components of the model chain. This may impact, in the future, the simulations presented in this article. Replacing ERA-40/ARPEGE by ERA-5 is under preparation and is expected to reduce temporal heterogeneities - although changes in observation data are likely to remain the dominant source of heterogeneity. It is also planned to expand the S2M reanalysis to lower-lying mountain ranges Vosges, Jura and Massif Central.

## 6    Conclusions

This study introduces and describes the latest release of the meteorological and snow cover reanalysis S2M covering the 63-year period from 1958 to 2021 for the French mountainous areas (French Alps, Pyrenees and Corsica). It includes the description of the different models and data used to produce this dataset, a comprehensive list of the parameters forming the dataset itself in its specific geometry and the technical access to the database. An evaluation of the simulation quality is provided by comparison to in-situ observations of snow depth as well as an overview of known and potential uses of this dataset, and a series of caveats associated with the use of this dataset. Yearly updates of this reanalysis will extend the period in the future and could lead to significant updates of the current S2M data by including model or input data modifications.

## 7    Data access

The S2M dataset is freely available on the AERIS data center on the following https://doi.org/10.25326/37. It is required that scientific publications using the S2M dataset mention in the acknowledgements: "The S2M data are provided by Météo-France - CNRS, CNRM, Centre d'Études de la Neige, through AERIS" and cite this article as reference and refer to the dataset as Vernay et al. (2022).

To access the data five fields are required :

- Versions : To date the most recent version is 2020.2

- Areas : Alpes (Alps), Pyrénées (Pyrenees), Corse (Corsica) massifs or Postes (stations). For the massif areas specify also the geometry type (flat or with 20° and 40° slopes)

- Products : "meteo" for meteorological variables (see section 3.1.2) and "snow" for snow cover and soil variables (see section 3.1.3)

- Begin year

- End year

Once the NetCDF files have been downloaded, it is possible to retrieve specific data by cross-checking the relevant metadata (see section 3.1.1).

*Acknowledgements.* This work was made possible thanks to the vision and pioneering work of people involved in the early steps of the SAFRAN - Crocus - MEPRA model chain (E. Brun, Y. Durand, E. Martin, G. Giraud and L. Mérindol).
The homogenised series of observation used for the evaluation of this dataset were made and provided by Brigitte Dubuisson (Météo-France, Direction de la Climatologie et des Services Climatiques, Toulouse, France).
CNRM/CEN is part of LabEx OSUG@2020.

| Massif number | Massif name | Department name | Associated department number |
|---|---|---|---|
| 1 | Chablais | Haute-Savoie | 74 |
| 2 | Aravis | Haute-Savoie | 74 |
| 3 | Mont-Blanc | Haute-Savoie | 74 |
| 3 | Bauges | Savoie | 73 |
| 5 | Beaufortin | Savoie | 73 |
| 6 | Haute-Tarentaise | Savoie | 73 |
| 7 | Chartreuse | Isère | 38 |
| 8 | Belledonne | Isère | 38 |
| 9 | Maurienne | Savoie | 73 |
| 10 | Vanoise | Savoie | 73 |
| 11 | Haute-Maurienne | Savoie | 73 |
| 12 | Grandes-Rousses | Isère | 38 |
| 13 | Thabor | Hautes-Alpes | 05 |
| 14 | Vercors | Isère / Drôme | 38 / 26 |
| 15 | Oisans | Isère | 38 |
| 16 | Pelvoux | Hautes-Alpes | 05 |
| 17 | Queyras | Hautes-Alpes | 05 |
| 18 | Devoluy | Hautes-Alpes | 05 |
| 19 | Champsaur | Hautes-Alpes | 05 |
| 20 | Parpaillon | Hautes-Alpes | 05 |
| 21 | Ubaye | Alpes de Haute-Provence | 04 |
| 22 | Haut-Var-Haut-Verdon | Alpes de Haute-Provence | 04 |
| 23 | Mercantour | Alpes-Maritimes | 06 |
| 40 | Cinto-Rotondo | Haute-Corse | 20 |
| 41 | Renoso-Incudine | Corse-du-Sud | 20 |
| 64 | Pays-Basque | Pyrénées-Atlantiques | 64 |
| 65 | Aspe-Ossau | Pyrénées-Atlantiques | 64 |
| 66 | Haute-Bigorre | Hautes-Pyrénées | 65 |
| 67 | Aure-Louron | Hautes-Pyrénées | 65 |
| 68 | Luchonnais | Haute-Garonne | 31 |
| 69 | Couserans | Ariège | 09 |
| 70 | Haute-Ariège | Ariège | 09 |
| 71 | Andorre | Andorre | 99 |
| 72 | Orlu-Saint-Barthélémy | Ariège | 09 |
| 73 | Capcir-Puymorens | Pyrénées-Orientales | 66 |
| 74 | Cerdagne-Canigou | Pyrénées-Orientales | 66 |

**Table 8.** List of massifs number, massifs names and associated department names and department numbers (used for the evaluation) included in the S2M reanalysis.

# 9 Appendix B

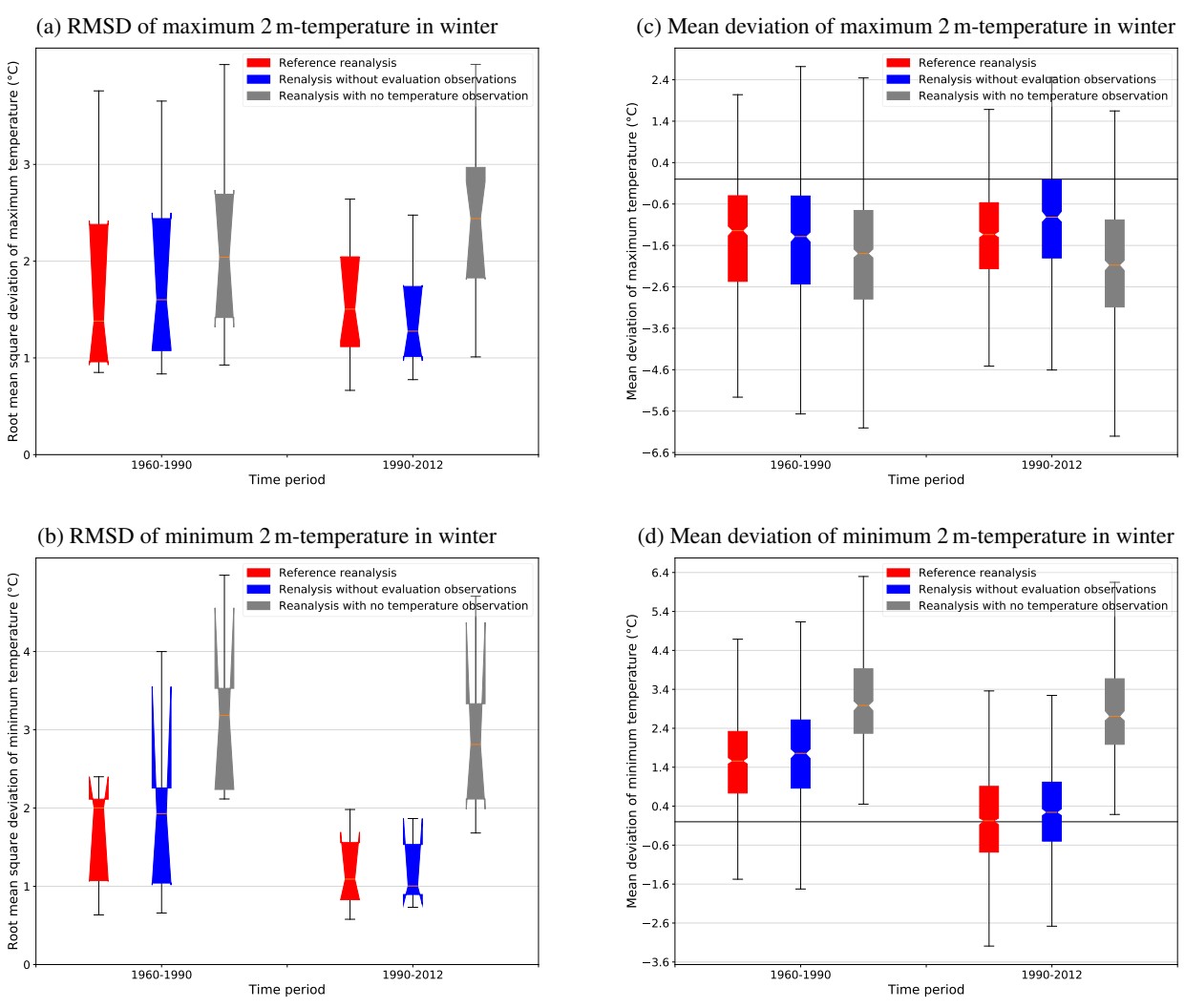

**Figure 16.** Same as Figure 10, but for winter (DJF) 2 m-temperatures only.

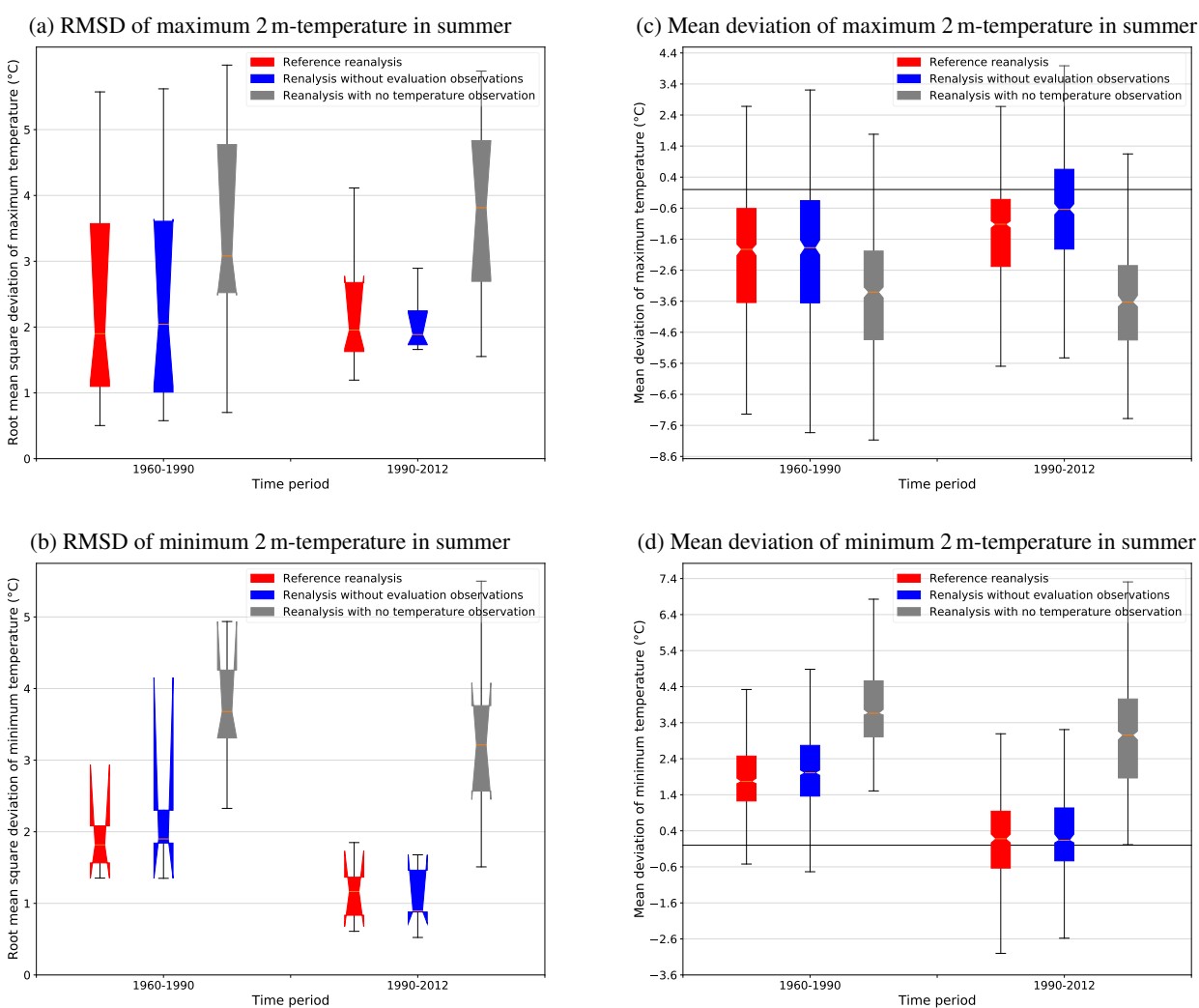

**Figure 17.** Same as Figure 10, but for summer (JJA) 2 m-temperatures only.

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
