# Peer review of "The S2M meteorological and snow cover reanalysis over the French mountainous areas: description and evaluation (1958 - 2021)"

_Earth System Science Data, 2021_

## Referee Comment (RC1)

Review of:

**The S2M meteorological and snow cover reanalysis over the Frenchmountainous areas, description and evaluation (1958 - 2020)**

By Matthieu Vernay et al.,

Major comments:

This is a paper about a very useful, unique and high-quality snow cover data set in mountain regions in Europe, which is relevant for researchers as well as various applications. The uniqueness of the data set lies in the full physically based information on snow cover properties. It is clearly worth publishing in ESSD. In fact, it contain even more than data on snow cover properties in a consistent manner (as snow cover is derived from a set of atmospheric driving variables), which are also part of the publication. Data are generally well described and presented. My comments are mainly related to the data evaluation/homogeneity and the description of the data set.

Data are derived from re-analysis simulations by well described unique models (SAFRAN for atmosphere and CROCUS and MEPRA for snow cover) originally forced by ERA-40 and ARPEGE. All models used are well described by peer-reviewed publications and are thus perfectly suited for the purpose of the application. Although the used approach of doing simulations at a spatial scale of mountain massifs results in some loss of spatial information, this is a suitable approach.

Data evaluation and data homogenmeity:

Driving data of re-analysis are both ERA-40 and ARPEGE. Additionally, e.g. precipitation data are used as guess for data assimilation which are based on AURELHY interpolation for period 1958-2017 and on ARPEGE thereafter (as ARPEGE is only available from 2017 onwards). All these changes cause (or at least could cause) inhomogeneities in the data series. Even if this inhomogeneities cannot be removed in the reanalysis, their effect should be discussed and if possible quantified (e.g. showing differences for precipitation between AURELHY and ARPEGE).

Interestingly, the temperature trend of the S2M reanalysis seems to be rather week over the period 1958-2020. This seems to be significantly weaker if compared to other data sets (as also mentioned in the paper). Are the trends described significant? It could be useful for the reader to see these differences in the trend curves in the figure (e.g. Fig. 5).

Data presentation:

The description of the S2M data set should generally be somewhat more detailed (see also the examples under "Minor Comments"). It should also be considered that non-meteorologists potentially want to use the data set and therefore the description of the metadata of the variables should be as detailed as possible.

Minor Comments:

Overall, be more specific with describing surface variables. E.g. is surface temperature 2m-temperature (which is used as term several times but not in all cases, is this something different then)?

Figures 3 and 4 as well as in the related text: Even though explained in the text, the terms "available" and "used" number of observations are misleading (as the number of used is higher compared to available). Suggest using other terms here.

Table at page 6 (which has no number, but should have one): Which variable is used for 300hPa? Which variable is measured at 10m? Which variable is measured at 1500m (again rel. humidity)? Be more specific as in table 2.

Figure 6: Why having a scale between 0 and 240 when values in the figure are much smaller?

4.2.3 is on trends of temperature and precipitation. However, related Figure 11 does not show trends but differences. Additionally, difference of total precipitation is in kg/m² which is not a unity of precipitation used frequently (suggest to change).

3.1.2 introduces the snow cover and soil variables. I could imagine that information on snow temperature could be useful as well. Please include the time reference (e.g. UTC) of variables.

Figure 8: Why has the trend of fraction of solid precipitation such strong increase for SON (and only for SON)? Additionally, be consistent between the figure and the figure caption by either showing trends or differences. Given that there is also an increase of precipitation and decrease in air temperature (for all elevations) for SON over 1960-90 to 1990-2020, I would expect also stronger increase for snow depth.

Figure 11: The S2M assimilation shows a clear elevation dependency of max. temperature change for 1960-90 vs. 1990-2020 for JJA, however not visible in station observations. Could be worth mentioning this EDW effect in the text.

4.3 evaluated snow depth observations: Given its relevance, it would be good to see how trends of snow depth are captured by the S2M reanalysis (compared to the independent station data).

---

## Author Comment (AC1)

**RC1**: 'Comment on essd-2021-249', Anonymous Referee #1, 11 Oct 2021

Review of:

**The S2M meteorological and snow cover reanalysis over the French mountainous areas, description and evaluation (1958 - 2020)**
By Matthieu Vernay et al.,

The authors would like to thank the reviewer for his/her remarks and suggestions to improve the quality of this manuscript.

Major comments:

This is a paper about a very useful, unique and high-quality snow cover data set in mountain regions in Europe, which is relevant for researchers as well as various applications. The uniqueness of the data set lies in the full physically based information on snow cover properties. It is clearly worth publishing in ESSD. In fact, it contain even more than data on snow cover properties in a consistent manner (as snow cover is derived from a set of atmospheric driving variables), which are also part of the publication. Data are generally well described and presented. My comments are mainly related to the data evaluation/homogeneity and the description of the data set.

Data are derived from re-analysis simulations by well described unique models (SAFRAN for atmosphere and CROCUS and MEPRA for snow cover) originally forced by ERA-40 and ARPEGE. All models used are well described by peer-reviewed publications and are thus perfectly suited for the purpose of the application. Although the used approach of doing simulations at a spatial scale of mountain massifs results in some loss of spatial information, this is a suitable approach.

Data evaluation and data homogeneity:

Driving data of re-analysis are both ERA-40 and ARPEGE. Additionally, e.g. precipitation data are used as guess for data assimilation which are based on AURELHY interpolation for period 1958-2017 and on ARPEGE thereafter (as ARPEGE is only available from 2017 onwards). All these changes cause (or at least could cause) inhomogeneities in the data series. Even if this inhomogeneities cannot be removed in the reanalysis, their effect should be discussed and if possible quantified (e.g. showing differences for precipitation between AURELHY and ARPEGE).

First please note that the available data do not allow a complete quantification of the impact of the guess transition from ERA40 to ARPEGE.
The impact of the temporal heterogeneity introduced by the different precipitation guess has been evaluated over one single season (2017-2018) by comparing a simulation made with precipitation guess based on the AURELHY analysis method to the reference one (with precipitation guess from ARPEGE). The comparison of the simulated snow depth mean deviation and root mean square deviations (RMSD) for these 2 configurations didn't show any significant impact on the performance of the system (see Figure 1 bellow).
Furthermore, the simulated annual precipitation amount with the precipitation guess based on the AURELHY analysis method seems to be slightly higher (about 3% on average) than those simulated with precipitation guess from ARPEGE: for the season 2017-2018 the average total precipitation over the 665 stations of the simulation with the guess based on the AURELHY analysis method and from ARPEGE respectively is 1507 mm (resp. 1464 mm) with accumulation ranging from 728 mm (resp. 675 mm) up to 3125 mm (resp. 3242 mm).
This impact is now mentioned in the revised manuscript in section 2.2.1 (line 138).

[Figure]

*Figure 1: Mean deviation (left) and RMSD (right) between the simulated and observed snow depths values on the 665 validation sites grouped by elevation range for two configurations of the system: with precipitation guess coming from the AURELHY analysis method (blue) and from ARPEGE (red)*

Interestingly, the temperature trend of the S2M reanalysis seems to be rather week over the period 1958-2020. This seems to be significantly weaker if compared to other data sets (as also mentioned in the paper). Are the trends described significant? It could be useful for the reader to see these differences in the trend curves in the figure (e.g. Fig. 5).

Details on simulated temperature trends and their significance have been added to the text related to Figure 5 (Line 306):
"The mean temperature trends simulated by the S2M reanalysis are +0.10°C per decade at 2700 m, +0.26°C per decade at 1800 m and +0.18°C per decade at 900 m. These trends are significant (all p-values of the trend slope significance are lower than 0.014)."

Data presentation:

The description of the S2M data set should generally be somewhat more detailed (see also the examples under "Minor Comments"). It should also be considered that non-meteorologists potentially want to use the data set and therefore the description of the metadata of the variables should be as detailed as possible.

More details have been added to the description of the data set. In particular a « metadata » section (section 3.1.1) describes the practical way to access specific simulation points and the new Table 2

summarizes the metadata which were previously not described. In addition section 7 (data access) has been extended to guide the data downloading.

Minor Comments:

Overall, be more specific with describing surface variables. E.g. is surface temperature 2m-temperature (which is used as term several times but not in all cases, is this something different then)?

The description of all surface variables (2m-temperature, 2m-humidity, 10m-wind) has been systematically specified, we thank the reviewer for pointing out this inaccuracy.

Figures 3 and 4 as well as in the related text: Even though explained in the text, the terms "available" and "used" number of observations are misleading (as the number of used is higher compared to available). Suggest using other terms here.

The terms "available" and "used" have been changed to "Available in massifs" and "Overall used" respectively in figure 3 and 4 as well as in the related text.

Table at page 6 (which has no number, but should have one): Which variable is used for 300hPa? Which variable is measured at 10m? Which variable is measured at 1500m (again rel. humidity)? Be more specific as in table 2.

We thank the reviewer for pointing out the missing table number. This table has been numbered (Table 1) and entirely rebuilt to be more specific as suggested.

Figure 6: Why having a scale between 0 and 240 when values in the figure are much smaller?

Figure 6 has been modified with a more consistent scale up to 120 cm, we thank again the reviewer for highlighting this issue.

4.2.3 is on trends of temperature and precipitation. However, related Figure 11 does not show trends but differences. Additionally, difference of total precipitation is in kg/m2 which is not a unity of precipitation used frequently (suggest to change).

To make the text easier to read, the term "trend" is used to describe the differences between two climatological periods. To clarify that use, the following sentence has been added at the beginning of Section 4.2.3:
"Here, climatological trends are defined by the difference between the mean of a variable over two 30-year long periods (e.g. 1990-2020 and 1960-1990)."

3.1.2 introduces the snow cover and soil variables. I could imagine that information on snow temperature could be useful as well. Please include the time reference (e.g. UTC) of variables.

The snow surface temperature is effectively in the data set, Table 4 has been modified to be more specific. The time reference of the variables has been included.

Figure 8: Why has the trend of fraction of solid precipitation such strong increase for SON (and only for SON)? Additionally, be consistent between the figure and the figure caption by either showing trends or differences. Given that there is also an increase of precipitation and decrease in

air temperature (for all elevations) for SON over 1960-90 to 1990-2020, I would expect also stronger increase for snow depth.

We thank the reviewer for this interesting remark concerning the interpretation of Figure 8. The strong increase of fraction of solid precipitation in autumn only and the comparatively low amplitude of the increase of the simulated snow depth may be explained by an averaging effect and the very short lifespan of snow on the ground during this season. We added this assumption to the text related to Figure 8 (Line 345). The caption of Figure 8 has also been modified to be consistent with the Figure showing differences.

Figure 11: The S2M assimilation shows a clear elevation dependency of max. temperature change for 1960-90 vs. 1990-2020 for JJA, however not visible in station observations. Could be worth mentioning this EDW effect in the text.

We added a mention of this divergence between the simulation and the observations in term of elevation dependency of max. temperature change between 1960-1990 and 1990-2020 in summer in the text at line 437:
"Figure 11 (b) shows that the simulated trends of maximum air temperature at 2 m in summer significantly increases with elevation up to about 1800 m a.s.l. However this elevation dependency of the simulated trend of maximum 2 m-temperature is not visible in station observations."

4.3 evaluated snow depth observations: Given its relevance, it would be good to see how trends of snow depth are captured by the S2M reanalysis (compared to the independent station data).

We thank the reviewer for this interesting suggestion. The evaluation of the simulated snow depth trends by the S2M reanalysis against the trends observed by independent station data is challenging due to the lack of long-enough observation series of snow depth. Most snow depth observations start in the 1990s and thus do not cover the 63 years of the S2M reanalysis. This temporal extent of available observations is a generalized issue over all European states (cf. Matiu et al, 2021). However Verfaillie et al.,2018 carried out a rough evaluation of the simulated snow depth at the long-term and independent observation site of Col de Porte which is one of the few stations with available observation data going back to 1960. Figure 2 of Verfaillie et al.,2018 indicates that the trend of simulated snow depth seems to match the observed one quite well at this specific point. It is especially worth noting that the simulation matches the observed decrease of snow depth since the 1990s.

Matiu, M., Crespi, A., Bertoldi, G., Carmagnola, C. M., Marty, C., Morin, S., Schöner, W., Cat Berro, D., Chiogna, G., De Gregorio, L., Kotlarski, S., Majone, B., Resch, G., Terzago, S., Valt, M., Beozzo, W., Cianfarra, P., Gouttevin, I., Marcolini, G., Notarnicola, C., Petitta, M., Scherrer, S. C., Strasser, U., Winkler, M., Zebisch, M., Cicogna, A., Cremonini, R., 620Debernardi, A., Faletto, M., Gaddo, M., Giovannini, L., Mercalli, L., Soubeyroux, J.-M., Sušnik, A., Trenti, A., Urbani, S., and Weilguni, V.: Observed snow depth trends in the European Alps: 1971 to 2019, 15, 1343–1382, https://doi.org/10.5194/tc-15-1343-2021, 2021.

Verfaillie, D., Lafaysse, M., Déqué, M., Eckert, N., Lejeune, Y., and Morin, S.: Multi-component ensembles of future meteorological and natural snow conditions for 1500 m altitude in the Chartreuse mountain range, Northern French Alps, The Cryosphere, 12, 1249–1271, https://doi.org/10.5194/tc-12-1249-2018, 2018.

---

## Author Comment (AC2)

**RC2**: 'Comment on essd-2021-249', Kristian Förster, 22 Nov 2021

Review of:

**"The S2M meteorological and snow cover reanalysis over the French mountainous areas, description and evaluation (1958 – 2020)"**

by Matthieu Vernay et al.

The authors would like to thank Kristian Förster for his remarks and suggestions to improve the quality of this manuscript.

This data paper presents a new snow cover reanalysis dataset for mountain areas in France. In contrast to other reanalysis products, the data is prepared for elementary areas, referred to as massifs. Delineating these elementary elements is based on the assumption that meteorological forcing is similar across each massif. For each of them, elevation bands and aspects are considered separately in order to summarize computational time and data. A set of models is used to downscale atmospheric reanalysis data (SAFRAN) and to predict snow cover (e.g., with Crocus) in the historic period 1958-2019. The paper comprehensively evaluates the accuracy of the data set (compared with station data), starting with meteorological data. Consequently, snow depth is evaluated. Finally, limitations are discussed in a well-balanced way, acknowledging uncertainties inherent in data and methods. I believe that this dataset is of great value for other researchers and I would recommend to publish this manuscript in ESSD, which is an ideal journal for this kind of research (data). I see only a few minor points that could be considered before publication:

General comments:

- ■ Besides snow depth, snow cover duration could be a very important quantity. It would be great to have another time series chart, showing how snow cover duration evolved over time (similar or to or as a sub-panel in Figure 5). It would be also interesting to see whether observed trends in snow cover duration are reproduced by your reanalysis (as addressed by Reviewer #1). Moreover, this would also demonstrate how the dataset could be used by others.

We thank the reviewer for this suggestion, we added a $5^{th}$ sub -panel in Figure 5 showing the temporal evolution of the simulated snow cover duration (SCD). The evaluation of this simulated SCD using surface observations is difficult since there are very few observations of SCD available over long time periods (most snow depth observations are carried out in ski resorts and stop before the total melting of snow and are thus useless in terms of SCD). Lopez-Moreno et al., 2020 carried out such an evaluation over the Pyrenees for the 2000–2017 period using MODIS products of SCD (Gascoin et al., 2015). Simulated SCD exhibits a good correlation with MODIS SCD (0.96) with a typical error of 20 days. The use of MODIS observations provides a comprehensive spatial coverage of snow cover duration observations but only for a short recent time period, preventing the evaluation of SCD long-term trends. However, work in progress at the CESBIO laboratory intends to use LandSat and SPOT archives to improve the length of time series back to 1985. Therefore, a more comprehensive evaluation of the trends of S2M snow cover durations will be done as soon as this product is available and could be expanded with lower resolution products such as Hüsler at al., 2014.  This perspective has been mentioned in the revised manuscript in section 5.2 (Line 536).

■ The definition of massifs as elementary elements for computation is a very interesting methodological approach of the paper, which could be interesting for future research. I found the description of this approach, however, rather vague. Here, I would expect a more comprehensive review of literature (e.g., summarizing areas with similar snow coverage is not so new, see, e.g., snow cover units etc.). Moreover, I was wondering how the variability of one of the massifs could look like. Maybe you could add a figure (appendix, supplement?) that shows the areas that are summarized in terms of aspect, elevation etc.

We added a review of literature of the semi-distributed modeling approach in Section 2.1. However, the definition of the massifs contours by Durand et al., 1999 was not based on an objective classification methodology but on an expert analysis of topography, difficult to describe in details (or to reproduce in another context).
Following the reviewer recommendation, we now illustrate the simplification of topography inside a massif by a new Figure 17 in appendix B.

■ When reviewing the nc files (meteo and snow, respectively), I was a bit confused about the dimensions: In the allslopes datasets, time series are provided for each number_of_points. Indeed, it would be possible to check for each number the combination of terrain characteristics (slope, elevation) but I couldn't find any further information (sorry, if I missed something). Even the shape files do not include any relation to the numbers and their associated terrain characteristics. I think that this could be better explained in the appendix / the repository. For users just interested in, say, south heading slopes in massif #1 at elevation above 2000 m, it would be helpful, if they could easily retrieve the relevant number(s).

We thank the reviewer for testing the access to the data set and to feed us back the difficulties he encountered. We added more details to the description of the data set. In particular a « metadata » section (section 3.1.1) describes the practical way to access specific simulation points and the new Table 2 summarize the metadata which were previously not described. In addition section 7 (data access) has been extended to guide the data downloading.

Specific comments / Technical comments:

• There is no reference to Table 5

The missing reference to Table 5 has been added on line 282 (Section 3.1.3)

• Please rewrite "precipitations" in the manuscript (the plural doesn't make sense in my opinion)

The word precipitation is now singular every time it appears, we thank the reviewer for pointing out this typo.

I am looking forward to your final revised paper!

Best wishes.

Gascoin, S., Hagolle, O., Huc, M., Jarlan, L., Dejoux, J.-F., Szczypta, C., Marti, R., and Sánchez, R.: A snow cover climatology for the Pyrenees from MODIS snow products, Hydrol. Earth Syst. Sci., 19, 2337–2351, https://doi.org/10.5194/hess-19-2337-2015, 2015.

Hüsler, F., Jonas, T., Riffler, M., Musial, J. P., and Wunderle, S.: A satellite-based snow cover climatology (1985–2011) for the European Alps derived from AVHRR data, The Cryosphere, 8, 73–90, https://doi.org/10.5194/tc-8-73-2014, 2014.

---

## Referee Report (RR1)

2nd Review

"The S2M meteorological and snow cover reanalysis over the French mountainous areas, description and evaluation (1958 - 2020)"

by Matthieu Vernay et al.

**Comments**

I really appreciate that the authors addressed snow cover duration in Figure 6d in their revised version of the manuscript! However, I still see some points that could be improved:

- The literature review has been improved and the statement on limitations is important here. Thank you! However, it still does not reflect early work in modelling snow cover with the concept of hydrological similar units, or more specifically "snow cover units" (SCU). I would recommend to replace the references in line 90 (in the track changes document) by more related earlier work referring to the SCU concept (even though more related to remote sensing of snow): Seidel et al. (1983) and Ehrler et al. (1997).

- The Section 3.1.1 on Metadata is not really helpful in its present form. I suggest to a few more details: What means "metadata" in the first column of Table 2? The definition of the dimension "number_of_points" is still missing. However, it would really help people to get in touch with your very useful dataset. I tried to follow your example (enumeration of details in Sect. 3.1.1). Please accept my apologies if I missed something but I still find it hard to put your example into practice. I took me some time and several lines of Python code to unravel your example by defining a selection which refers to the dimension "number_of_points". I think a few more technical details would be helpful to get started with the data. Maybe you could add a few lines of example code to the appendix or at least some pseudocode to better explain data usage (see my example, which indeed could be improved)? Maybe there are better ways to apply your example I am not aware of ☺

The paper has a very high quality and I would suggest technical revisions to better reflect the literature on hydrological similar units and to improve the details on the nc files.

I am looking forward to your final published paper!

Best wishes.

**References**

Ehrler, C., Seidel, K., & Martinec, J. (1997). Advanced analysis of snow cover based on satellite remote sensing for the assessment of. Remote Sensing and Geographic Information Systems for Design and Operation of Water Resources Systems, IAHS Publication (242), 93.

Seidel, K., Ade, F., & Lichtenegger, J. (1983). Augmenting LANDSAT MSS data with topographic information for enhanced registration and classification. IEEE transactions on geoscience and remote sensing, (3), 252-258.

**Code listing**

```python
import xarray as xr
import numpy as np
import matplotlib.pyplot as plt

**open nc file**
ncdata = xr.open_dataset('PRO_2019080106_2020080106.nc')

**retrieve data for diminesions / variables**
index = ncdata.coords['time'].to_dataframe()['time'].values
var_swe = ncdata.variables['SWE_1DY_ISBA']
var_slope = ncdata.variables['slope'].data
var_aspect = ncdata.variables['aspect'].data
var_massif = ncdata.variables['massif_num'].data
var_zs = ncdata.variables['ZS'].data

**select data according to Sect. 3.1.1**
selection = np.squeeze(np.where((var_zs>=1800)&(var_massif==1)&(var_aspect==180.)), axis=0)

**define labels for different eleveation zones**
labels = [var_zs[si] for si in selection]

**plot data & quit**
plt.contourf(labels,index,var_swe[:,selection].data)
plt.colorbar()
ncdata.close()
```

---

## Author Response (AR2)

2nd Review

"The S2M meteorological and snow cover reanalysis over the French mountainous areas, description and evaluation (1958 – 2020)"

by Matthieu Vernay et al.

**Comments**

I really appreciate that the authors addressed snow cover duration in Figure 6d in their revised version of the manuscript! However, I still see some points that could be improved:

- The literature review has been improved and the statement on limitations is important here. Thank you! However, it still does not reflect early work in modelling snow cover with the concept of hydrological similar units, or more specifically "snow cover units" (SCU). I would recommend to replace the references in line 90 (in the track changes document) by more related earlier work referring to the SCU concept (even though more related to remote sensing of snow): Seidel et al. (1983) and Ehrler et al. (1997).

We thank Kristian Förster for suggesting additional references concerning snow cover units, they have been added on line 90.

- The Section 3.1.1 on Metadata is not really helpful in its present form. I suggest to add few more details: What means "metadata" in the first column of Table 2? The definition of the dimension "number_of_points" is still missing. However, it would really help people to get in touch with your very useful dataset. I tried to follow your example (enumeration of details in Sect. 3.1.1). Please accept my apologies if I missed something but I still find it hard to put your example into practice. I took me some time and several lines of Python code to unravel your example by defining a selection which refers to the dimension "number_of_points". I think a few more technical details would be helpful to get started with the data. Maybe you could add a few lines of example code to the appendix or at least some pseudocode to better explain data usage (see my example, which indeed could be improved)? Maybe there are better ways to apply your example I am not aware of ☺

We thank again Kristian Föster for testing the manipulation of the S2M dataset. More details have been added on section 3.1.1, especially the first column of Table 2 has been renamed "Geometry variables" to be more specific. The provided python code has been slightly modified in order to retrieve simulation data corresponding to the example in section 3.1.1 (plot of the full temporal series and extract the value for one specific date) and we added few lines of code to deal with the massif shapefiles in order to plot maps. The resulting code has been added as supplement.

The paper has a very high quality and I would suggest technical revisions to better reflect the literature on hydrological similar units and to improve the details on the nc files. I am looking forward to your final published paper!

Best wishes.